# Pro-oxidant response and accelerated ferroptosis caused by synergetic Au(I) release in hypercarbon-centered gold(I) cluster prodrugs

Kui Xiao[1,6], Niyuan Zhang[2,6], Feifei Li[3,6], Dayong Hou ⊕[2,4,5,6], Xiaoyi Zhai[1], Wanhai Xu[4,5] ✉, Gelin Wang[3] ✉, Hao Wang[2] ✉ & Liang Zhao ⊕[1] ✉

Medicinal applications of gold complexes have recently attracted attention due to their innovative antitumor mechanisms. In this work, two hypercoordinated carbon-centered gold clusters **PAA4** and **PAA5** are quantitatively synthesized by an intramolecular 6-*exo-dig* cyclization of polymetalated precursors. The on-bench and in vitro experimental studies demonstrate that the characteristic hypercarbon-tetragold(I) multi-center bonding in **PAA4** and **PAA5** not only guarantees their stability under common physiological conditions, but also facilitates a glutathione (GSH)-triggered prompt and synergetic release of active Au(I) ions in the GSH-overexpressed and acidic microenvironment of human bladder cancer EJ cells. The instantly massive release of coordination unsaturated Au(I) ions causes the efficient inhibition of thioredoxin reductases and then induces a rapid pro-oxidant response, consequently causing the occurrence of accelerated ferroptosis of EJ cells. As a result, these hypercarbon-centered gold(I) cluster prodrugs show high cytotoxicity to bladder cancer cell lines and thus exhibit a significant inhibition effect towards bladder tumors in vivo. Correlation of the synergetic domino dissociation of carbon-polymetal multi-center bonding in metal clusters with the accelerated ferroptosis of cancer cells provides a strategy for metalloprodrugs and opens a broader prospect for the biological application of metal cluster compounds.

Ferroptosis has attracted much recent attention as this non-apoptotic programmed cell death (PCD) process provides an alternative strategy in tumor suppression[1–4]. In particularly, some cancer cells resistant to conventional therapies are vulnerable to ferroptosis, thus making ferroptosis-associated drugs great potential in tumor therapy[5,6]. Generally, the main intrinsic pathway of ferroptosis relies on the activation

[1]Key Laboratory of Bioorganic Phosphorus Chemistry and Chemical Biology (Ministry of Education), Department of Chemistry, Tsinghua University, Beijing 100084, China. [2]CAS Key Laboratory for Biomedical Effects of Nanomaterials and Nanosafety, CAS Center for Excellence in Nanoscience, National Center for Nanoscience and Technology (NCNST), Beijing 100190, China. [3]School of Pharmaceutical Sciences, Tsinghua-Peking Joint Center for Life Sciences, Tsinghua University, Beijing 100084, China. [4]Department of Urology, the Fourth Hospital of Harbin Medical University, Heilongjiang Key Laboratory of Scientific Research in Urology, Harbin 150001, China. [5]NHC Key Laboratory of Molecular Probes and Targeted Diagnosis and Therapy, Harbin Medical University, Harbin 150001, China. [6]These authors contributed equally: Kui Xiao, Niyuan Zhang, Feifei Li, Dayong Hou. ✉e-mail: xuwanhai@hrbmu.edu.cn; gelinwang@tsinghua.edu.cn; wanghao@nanoctr.cn; zhaolchem@mail.tsinghua.edu.cn

by imbalanced redox homeostasis and the subsequent accumulation of lipid peroxides[4]. Thus, the inhibition of redox-active enzymes toward the regulation of antioxidant systems often account for the initiation of ferroptosis[5,7]. Gold anticancer drugs have been found efficiently suppressing chalcogen-containing enzymes in antioxidant systems (e.g. thioredoxin reductase TrxR and glutathione reductase) and consequently altering cellular redox environment[8–11]. Recent investigation has revealed that a high dose of mononuclear gold(I) drug can induce ferroptosis through the inhibition of the TrxR activity[12]. The prerequisite high dose of gold(I) drugs is ascribed to the keen competition between effective enzyme binding and off-targeted binding with thiol ligands such as glutathione (GSH)[8]. However, a high dose of administration is often accompanied with systemic toxicity towards normal tissues[13]. In this regard, a prodrug strategy capable of maximizing the intracellular efficient concentration of gold(I) drugs and simultaneously minimizing side effects is highly desired with an aim at medical application of gold(I) drugs via in vivo ferroptosis-based tumor therapy.

Hypercarbon-centered gold(I) clusters feature a coordination-oversaturated [RC-Au$^I_n$] kernel, in which the central carbon atom is hyper-coordinated with four or more Au(I) atoms[14–17]. The presence of multi-center bonding and multifold aurophilic interactions enable the hypercarbon-polygold entities with superb stability[18,19], which helpfully promotes the intracellular on-targeted drug delivery. Moreover, the multi-center bonding is potentially featured with a consecutive bond dissociation process, wherein the cleavage of a single carbon–gold bond in the [RC-Au$^I_n$] moiety caused by strong coordinative thiol ligands (e.g. GSH) may induce a synergistic bond rupture and then lead to a domino release of coordination-unsaturated Au(I) ions (Fig. 1). This thiol-caused domino-type Au(I) release probably reverses the deleterious GSH-binding for common gold(I) anticancer drugs to become an advantageous pathophysiological trigger for gold(I) cluster-based prodrugs.

In this work, we synthesize two methide carbon-centered poly-nuclear Au(I) clusters **PAA4** and **PAA5** via an in situ cyclization reaction of polymetalated precursors. These hypercarbon-centered gold(I) clusters show good stability in cell culture medium, while they are activated by GSH to trigger a consecutive release of highly active [AuPPh$_3$]$^+$ ions in solution. In particular, the release process is more rapid in human bladder cancer EJ cells due to their GSH-overexpressed and acidic microenvironment. The sudden surge of coordination unsaturated Au(I) ions results in a kinetical enhancement of reactive oxygen species (ROS) by inhibiting the activity of TrxR and triggering a pro-oxidant response, conse-quently inducing a rapid lipid peroxidation and then accelerating the ferroptosis process of EJ cells (Fig. 1). Moreover, in vivo ther-apy investigation in mice bladder tumor models demonstrates excellent anti-tumor performance and low systemic toxicity of these

hypercarbon-centered gold(I) clusters. This work demonstrates a strategy for metallo-prodrugs and realizes a ferroptosis-based ther-apy for cancer cells based on organometallic kinetics.

## Results

### Synthesis and characterization of hypercarbon-centered orga-nogold(I) clusters PAA4 and PAA5

The classical synthesis of hypercoordinated carbon-centered metal compounds mainly depends on the use of specific acidic precursors such as ylides, nitriles, and methyl ketones, which are subject to strong bases to generate carbon polyanions by deprotonation. Besides the limited structural diversity of precursors, such multiple direct deprotonation often causes unwanted side reactions. We herein rely on an intramolecular cyclization of polyaurated intermediates[20–22] to in situ generate hypercarbon-centered gold clusters. As shown in Fig. 2a, the NH$_2$ group in the precursor 2′-((tri-methylsilyl)ethynyl)-[1,1′-biphenyl]−2-amine (TEBA) was activated by [(AuPPh$_3$)$_3$(μ$_3$-O)](BF$_4$) ([Au$_3$O]) to produce a highly nucleophilic [NAu$_3$] moiety. This process was accompanied with the desilylation and metalation of the [C≡C-SiMe$_3$] group by fluoride and [AuPPh$_3$] (BF$_4$), respectively. Finally, the doubly activated TEBA underwent a favored 6-exo-dig cyclization to generate a hypercarbon-centered tetranuclear cluster complex **PAA4** in a high isolated yield of 79%. In view of the nitrogen coordination vacancy site on the resulting 6-methylphenanthridine in **PAA4**, we purposefully added one more equivalent [AuPPh$_3$](BF$_4$) to quantitatively acquire a penta-aurated complex **PAA5**.

X-ray crystallographic analysis reveals that the methide carbon atom of the 6-methylphenanthridine in **PAA4** lies 0.76 Å above a coplanar gold square (Au···Au: 2.811(1)−2.875(1) Å), while the prototype carbon in **PAA5** bonds to a nest-like Au$_5$ unit (Au···Au: 2.821(1)−3.211(1) Å) via a μ$_5$-C1,N1-η$^4$,η$^1$ mode (Fig. 2b). Clearly, the attachment of one more gold atom onto the square [C-Au$^I_4$] moiety via the extra nitrogen coordination site leads to the cluster distortion in **PAA5** due to remarkable steric hindrance among the crowded PPh$_3$ ligands (Fig. 2b). The C−Au distances (2.134(10)−2.210(10) Å) in **PAA4** and **PAA5** are comparable to typical carbon-polygold multi-centered bonds in the reported organogold(I) clusters[23–26]. **PAA4** and **PAA5** keep the [C-Au$^I_4$] and [C,N-Au$^I_5$] core structures intact in solution as evidenced by electron-spray ionization mass spectroscopy (ESI-MS) and nuclear magnetic resonance (NMR) (Supplementary Figs. S1−S7). The well-resolved high-resolution peaks in ESI-MS can be assigned as the species arising from **PAA4** and **PAA5** minus one or two tetrafluoroborate anions (Supplementary Figs. S6 and S7). The $^{31}$P NMR spectrum of **PAA4** reveals only one resonance peak at 31.3 ppm corresponding to four equivalent [AuPPh$_3$] units around the hypercarbon atom (Supplementary Fig. S3). In contrast, two resonance peaks at 31.1 and 28.1 ppm appear in the $^{31}$P NMR spectrum of **PAA5**, which should be

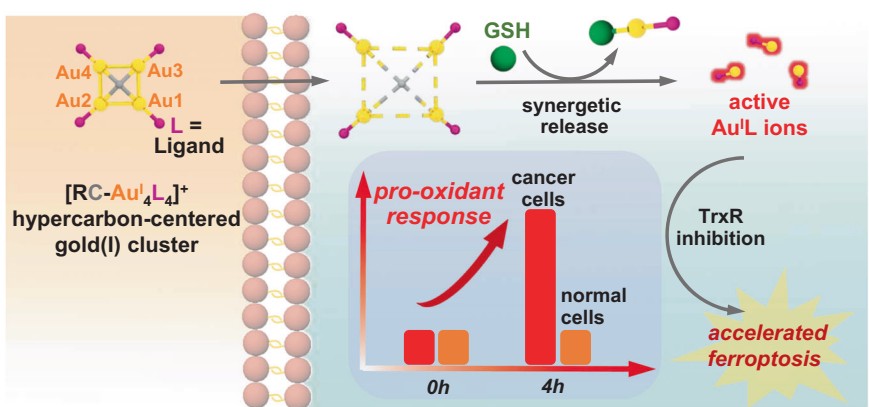

**Fig. 1 | Schematic of pharmacological mechanisms of hypercarbon-centered gold(I) cluster prodrugs.**

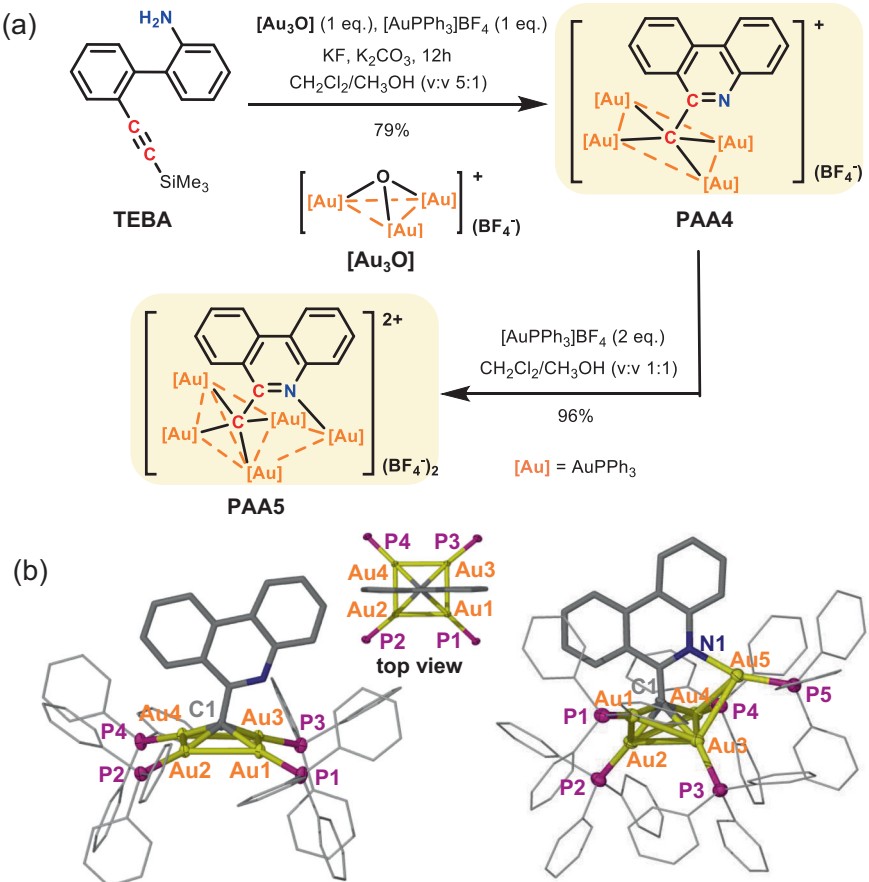

**Fig. 2 | Synthesis and crystal structures of PAA4 and PAA5. a** Synthetic procedures for **PAA4** and **PAA5**. **b** Crystal structures of **PAA4** (left) and **PAA5** (right). Hydrogen atoms and tetrafluoroborate counter anions are omitted for clarity. Selected bond lengths (Å) and angles (°) in **PAA4**: Au1-Au2 2.875(1), Au2-Au3 2.811(1), C1-Au1 2.146(4), C1-Au2 2.152(4), ∠Au2-Au1-Au3 89.79(1), ∠Au1-Au2-Au4

90.19(1). **PAA5**: Au1-Au2 2.954(1), Au1-Au3 2.821(1), Au2-Au3 3.211(1), Au2-Au4 2.829(1), Au3-Au4 2.892(1), Au3-Au5 3.016(1), Au4-Au5 2.989(1), C1-Au1 2.179(9), C1-Au2 2.134(10), C1-Au3 2.165(11), C1-Au4 2.210(10), N1-Au5 2.114(9), ∠Au1-Au2-Au4 96.29(1), ∠Au1-Au3-Au4 97.88(2), ∠Au2-Au1-Au3 67.51(1), ∠Au2-Au4-Au3 68.28(1).

assigned to the [AuPPh₃] units attached on the carbon and nitrogen atoms, respectively (Supplementary Fig. S5).

With an aim at the biological application of these organogold(I) clusters, the stability of hypercarbon-centered gold(I) clusters was evaluated in common solvents and cell culture conditions. First, we observed negligible signal change in the ¹H NMR monitoring on the solution sample of **PAA4** upon heated up to 353 K or exposure to a mixed solution of dimethyl sulfoxide (DMSO) and water for 24 h (Supplementary Figs. S8 and S9). This result verifies the good thermal and chemical stability of **PAA4**. Moreover, upon dissolving **PAA4** in phosphate buffer saline (PBS) solutions with a wide pH range of 4.9–8.7 or in a widely used Dulbecco's modified eagle medium (DMEM) containing a highly concentrated glucose and glutamine (Supplementary Figs. S10 and S11), the UV–Vis spectra remain unchanged within 1 h. In addition, the ¹H NMR investigation reveals that **PAA4** shows no reaction with the disulfide compound cystine, a major component of the cell culture medium (Supplementary Fig. S12). The good stability of **PAA5** was also verified by ¹H NMR studies and UV–Vis spectra (Supplementary Figs. S9, S12, and S13). The excellent stability of **PAA4** and **PAA5** provides a structural basis for their potential prodrug application.

### Synergetic Au(I) release of PAA4 and PAA5 activated by GSH

The microenvironment of tumor cells is significantly distinct from normal cells in the aspect of such as an elevated level of GSH and acidity[27,28]. Therefore, the anticancer efficiency of common gold(I) drugs often suffers from a severe thiol-binding competition between

targeted enzymes and the off-targeted GSH[8]. We next conducted the reactivity studies of **PAA4** and **PAA5** with GSH based on the ¹H NMR and UV–Vis spectroscopic monitoring. Firstly, the mixing of **PAA4** with GSH in an aqueous solution leads to the disappearance of three characteristic ¹H NMR doublet peaks (10.40, 8.63, and 8.52 ppm) corresponding to the hydrogen atoms (Ha, Hd, and He) on the phenanthridine ring of **PAA4**. Notably, even less than one equivalent GSH can cause the complete decomposition of **PAA4** (Fig. 3b), implying the occurrence of a blasting-type carbon–gold bond rupture. ESI-MS confirms the formation of 6-methylphenanthridine (MPA) via a three-step protodeauration triggered by less than one equivalent GSH (Fig. 3a and Supplementary Fig. S14). To further identify the in situ generated Au(I) species derived from the reaction of **PAA4** and GSH, we selected a probe compound 8-((trimethylsilyl)ethynyl)naphthalen-1-amine (ENA), which has been previously found undergoing a prompt reaction with the coordination unsaturated [AuPPh₃](BF₄) to produce a cyclized mononuclear gold complex [(2-methylbenzo[cd]indole)AuPPh₃](BF₄) (MBIA)[21] (Fig. 3a and Supplementary Fig. S15). Upon the addition of ENA into the reaction mixture of **PAA4** and GSH (Fig. 3c,d), the fluorescence emission at 500 nm corresponding to ENA gradually vanished, which was accompanied with the increase of the absorption of MBIA at 510 nm. More significantly, addition of acid HBF₄ (pH =3.5) further promotes the reactivity of **PAA4** towards ENA (Fig. 3c, inset). The acceleration effect of acid is further verified by the more rapid reaction between GSH and **PAA4** or **PAA5** with HBF₄ addition (Supplementary Fig. S16b). These results verify that the [C-Au¹₄] moiety in **PAA4** can be activated by GSH and

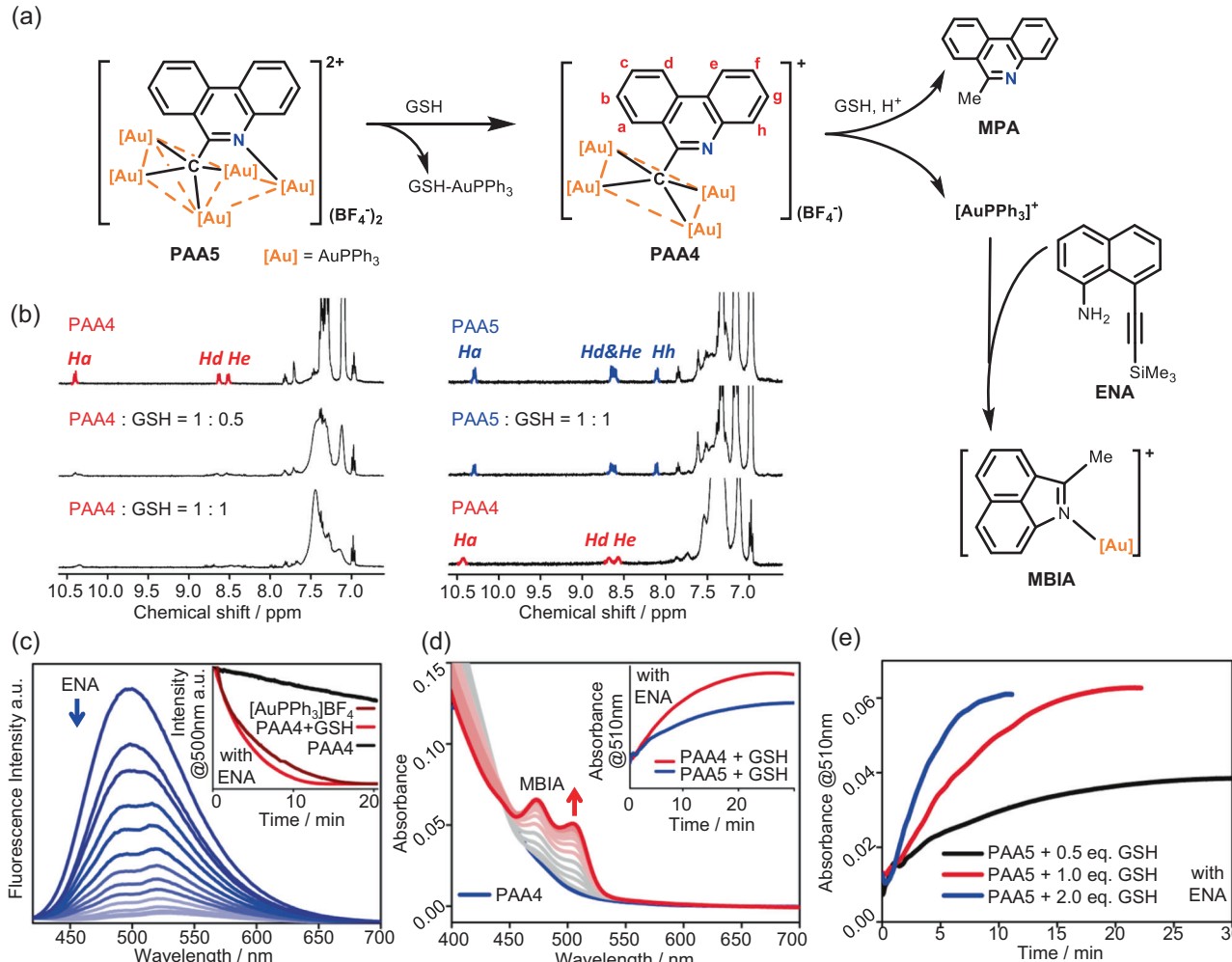

**Fig. 3 | Reactivity studies of PAA4 and PAA5 with GSH. a** Schematic reaction procedures of **PAA4** and **PAA5** with GSH and the ENA-to-MBIA transformation triggered by active Au(I) species. **b** $^1$H-NMR spectra monitoring (DMSO-d$_6$:D$_2$O = 9:1, 298 K) on the reaction mixture of **PAA4**/GSH (left) and **PAA5**/GSH (right) in different ratios. **c** Fluorescence spectrum monitoring ($\lambda_{ex}$ = 380 nm) on the reaction mixture of **PAA4**:ENA:GSH = 1:0.5:1 ($c_{PAA4}$ = 3.3 × 10$^{-5}$ M, MeOH:H$_2$O = 1:1, 298 K) in the presence of HBF$_4$ ($c_{HBF4}$ = 1.0 × 10$^{-4}$ M). Inset: Fluorescence variation curves at 500 nm ($\lambda_{ex}$ = 380 nm) corresponding to the decrease of ENA ($c_{ENA}$ = 1.6 × 10$^{-5}$ M) upon its reaction with **PAA4** ($c_{PAA4}$ = 3.3 × 10$^{-5}$ M), [AuPPh$_3$]BF$_4$ (1.3 ×

10$^{-4}$ M), or the mixture of **PAA4**:GSH = 1:1 ($c_{PAA4}$ = 3.3 × 10$^{-5}$ M). **d** UV–Vis spectrum monitoring of the reaction mixture of **PAA4**:ENA:GSH = 1:4:0.5 ($c_{PAA4}$ = 1.0 × 10$^{-5}$ M, DMSO:H$_2$O = 1:1, 298 K). Inset: UV–Vis spectrum monitoring of **MBIA** generated in the reaction mixtures of **PAA4**:ENA:GSH = 1:4:0.5 and **PAA5**:ENA:GSH = 1:4:0.5 ($c_{PAA4}$ = $c_{PAA5}$ = 1.0 × 10$^{-5}$ M, DMSO:H$_2$O = 1:1, 298 K). **e** Absorbance variation curves at 510 nm corresponding to the reaction mixtures of **PAA5**/GSH/ENA in different ratios ($c_{PAA5}$ = 1.0 × 10$^{-5}$ M, $c_{ENA}$ = 4.0 × 10$^{-5}$ M, DMSO:H$_2$O = 1:1, 298 K). ($n$ = 2 independent experiments).

facilitated by acid to consecutively release active Au(I) species and thus facilitate the ENA-to-MBIA transformation. In addition, the ENA-to-MBIA transformation can be accelerated along with the concentration increase of GSH (Fig. 3e and Supplementary Fig. S17). Moreover, the role of GSH in the acceleration of the [AuPPh$_3$]$^+$ release from **PAA4** and **PAA5** are further verified to be in line with the concentration of GSH from 50-to-100 equivalents (Supplementary Fig. S16). The ENA-to-MBIA transformation also occurs in the reaction mixture of **PAA5** and GSH (Supplementary Fig. S18), but in a relatively slow way (Fig. 3d, inset). Such delay of reaction between **PAA5** and GSH has also been verified via the smaller reaction rates comparing to those of **PAA4** and GSH (Supplementary Fig. S17d). The in situ $^1$H NMR monitoring substantiates that **PAA5** has to experience a degradation to produce **PAA4** upon its reaction with the first equivalent GSH (Fig. 3b). These on-bench experimental results indicate that GSH can serve as a physiological trigger to facilitate the consecutive release of active Au(I) species from the parent hypercarbon-centered gold(I) clusters, finally enabling **PAA4** and **PAA5** as a potential metallo-prodrug[28].

## Ferroptosis induced by PAA4 and PAA5 within bladder cancer cells

In order to prove the potential prodrug role of the hypercarbon-centered gold clusters in antitumor treatment, we first tested the viability of cancer and normal cells treated by **PAA4** and **PAA5**. Human bladder cancer EJ cells were selected because of their remarkable GSH over-expression and acidic microenvironment, which may facilitate the protodeauration of **PAA4** and **PAA5** and causes consecutive Au(I) release as shown in the above on-bench experiment[29,30]. The human umbilical vein endothelial cell (HUVEC) line was applied as a contrast normal cell example. In addition, Ph$_3$PAuCl (**PA1**) was applied as a contrast example of coordination-saturated mononuclear gold(I) species. First, we confirm that the organic product MPA, which can be produced by the full protodeauration of **PAA4** and **PAA5**, has no significant cytotoxicity in EJ and HUVEC cell lines (Supplementary Fig. S19). **PAA4**, **PAA5**, and **PA1** were then respectively incubated with EJ and HUVEC cells for 24 h. As shown in Fig. 4a, upon the treatment by 1.5 μM **PAA4** and **PAA5**, the cell viability of EJ cells decreases 57 and 55%, respectively, relative to the HUVEC cells. This biased cell mortality

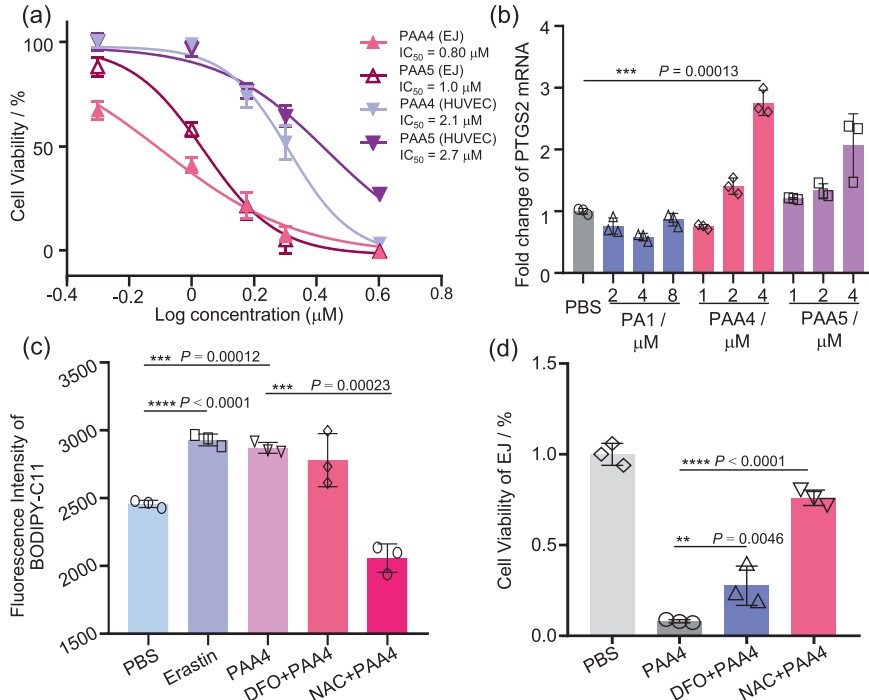

**Fig. 4 | In vitro antitumor activity evaluation of PAA4 and PAA5. a** Cell viability assay of EJ and HUVEC cells treated with **PAA4** (IC50 0.8 μM for EJ cells and 2.1 μM for HUVEC cells) and **PAA5** (IC50 1.0 μM for EJ cells and 2.7 μM for HUVEC cells) for 24 h. **b** Changes of PTGS2 mRNA in EJ cells treated by **PAA4**, **PAA5**, and **PA1** for 6 h. **c** Fluorescence intensity of lipid peroxidation labeled by BODIPY-C11 in EJ cells. DFO (100 μM) or NAC (3.0 mM) was pretreated 2 h before the **PAA4** (4.0 μM, 6 h incubation) treatment. **d** Rescue effect of DFO (100 μM) or NAC (3.0 mM) on the PAA4-treated (4.0 μM) EJ cell lines. DFO (100 μM) or NAC (3.0 mM) was pretreated 2 h before **PAA4** (4.0 μM, 6 h incubation) treatment. Asterisks (*) denote the statistical significance: $0.001< ** P \leq 0.01$, $0.0001< *** P \leq 0.001$, $**** P \leq 0.0001$, $P$ values were performed with one-way ANOVA followed by post hoc Tukey's test. Data were expressed as mean ± SD ($n = 3$ independent experiments examined over triplicates). Source data are provided as a Source Data file.

demonstrates an antitumor effect of **PAA4** and **PAA5** in the EJ cell line. Notably, the mononuclear gold(I) compound **PA1** contrarily exhibits low cytotoxicity towards the EJ cells, half maximal inhibitory concentration (IC50) 6.0 μM (Supplementary Fig. S20). This result indicates that active Au$^{I}$ species arising from the GSH-triggered dissociation of **PAA4** and **PAA5** should account for the antitumor performance in the EJ cell lines.

Next, we found that **PAA4** and **PAA5** cause a significant acceleration death of EJ cells relative to HUVEC cells (Supplementary Fig. S21). We subsequently conducted mechanistic studies to clarify the PCD type accounting for such acceleration, which focused on two different pathways, apoptosis and ferroptosis. Generally, the widely recognized PCD type for gold(I) anticancer drugs is apoptosis[9]. Therefore, we tested the cleavage of poly (ADP-ribose) polymerase (PARP), regulation of caspase 3 and expression of Bcl-2 Associated X-protein (Bax), symbols for apoptosis. In contrast to the significant activation of apoptosis in the PA1-treated HUVEC cell lines, no apoptosis biomarker was observed in both EJ and HUVEC cell lines under the treatment with **PAA4** and **PAA5** (Supplementary Fig. S22). Furthermore, the addition of the apoptosis inhibitor benzyloxycarbonyl-Val-Ala-Asp-(OMe)-fluoromethyl ketone (ZVAD) cannot alleviate the cell mortality in both EJ cells incubated with **PAA4** and **PAA5**. These results suggest that **PAA4** and **PAA5** hardly cause apoptosis. On the other hand, we observed the increase of the ferroptosis marker gene prostaglandin-endoperoxide synthase 2 (PTGS2)[31] after the treatment by **PAA4** and **PAA5** (Fig. 4b). Notably, the **PAA4**-treated sample gave rise to 2.8-fold over-expression of PTGS2 mRNA relative to the blank group. The up-regulation of PTGS2 protein was also verified in **PAA4**-treated EJ cells (Supplementary Fig. S23). Moreover, ACSL mRNA, another biomarker of ferroptosis generation[32], also gave a slight increase after treated by **PAA4** and **PAA5** (Supplementary Fig. S24c). To further verify the occurrence of the ferroptotic cell death in the

PAA4- and PAA5-treated EJ cancer cells, we monitored the concentration of lipid peroxides, which is a hallmark of ferroptosis, by tracking the fluorescence of a lipid peroxidation sensor BODIPY-C11. As shown in Fig. 4c, after incubated with 4.0 μM **PAA4** for 6 h, a rapid increase of lipid peroxides in the EJ cells was observed, which is well-matched with the over-expression of PTGS2. As a positive control, the well-defined ferroptosis inducer Erastin[3] also causes enhancement of lipid peroxides in EJ cells. Moreover, the addition of ferroptosis inhibitor iron chelator deferoxamine (DFO) or antioxidant N-Acetyl-L-cysteine (NAC) also leads to the decrease of the cell mortality ratio (Fig. 4d and Supplementary Fig. S21), which is accompanied with a simultaneous restraint of lipid peroxidation (Fig. 4b). In contrast, **PA1** is incapable of inducing lipid peroxidation under the same condition (Supplementary Figs. S25), which is consistent with its low cytotoxicity relative to the clusters. Here is no significant PTGS2 protein expression or lipid peroxidation found in **PAA4**- and **PAA5**-treated HUVEC cells (Supplementary Figs. S23 and S26). These results jointly demonstrate that the accelerated EJ cancer cell death is ascribed to ferroptosis.

We then analyzed the pharmacological pathways in vitro by details (Fig. 5a). Firstly, inductively coupled plasma mass spectrometry (ICP-MS) revealed the up-take amount of **PAA4** and **PAA5** by EJ cells as $2.26 \pm 0.02 \times 10^{-15}$ and $1.98 \pm 0.03 \times 10^{-15}$ mol/cell within 24 h, respectively, corresponding to an up-take ratio of 30% and 26% upon a dose of 1.5 μM in cell culture medium. Therein, about 10% gold clusters were found at mitochondria according to ICP-MS results (Supplementary Fig. S27). It is worth noting that the electroneutral compound **PA1** (6.0 μM) gives rise to a lower cellular up-take than the cationic **PAA4** (1.5 μM) within 4 h ($4.46 \pm 0.23 \times 10^{-16} < 6.00 \pm 0.06 \times 10^{-16}$ mol/cell, Fig. 5b). In view of the reported cellular up-take of gold clusters by means of clathrin-mediated endocytosis and macropinocytosis[33], we deduce that the accumulation of the PPh$_3$-protected Au clusters inside mitochondria should be ascribed to the lipophilic and cationic

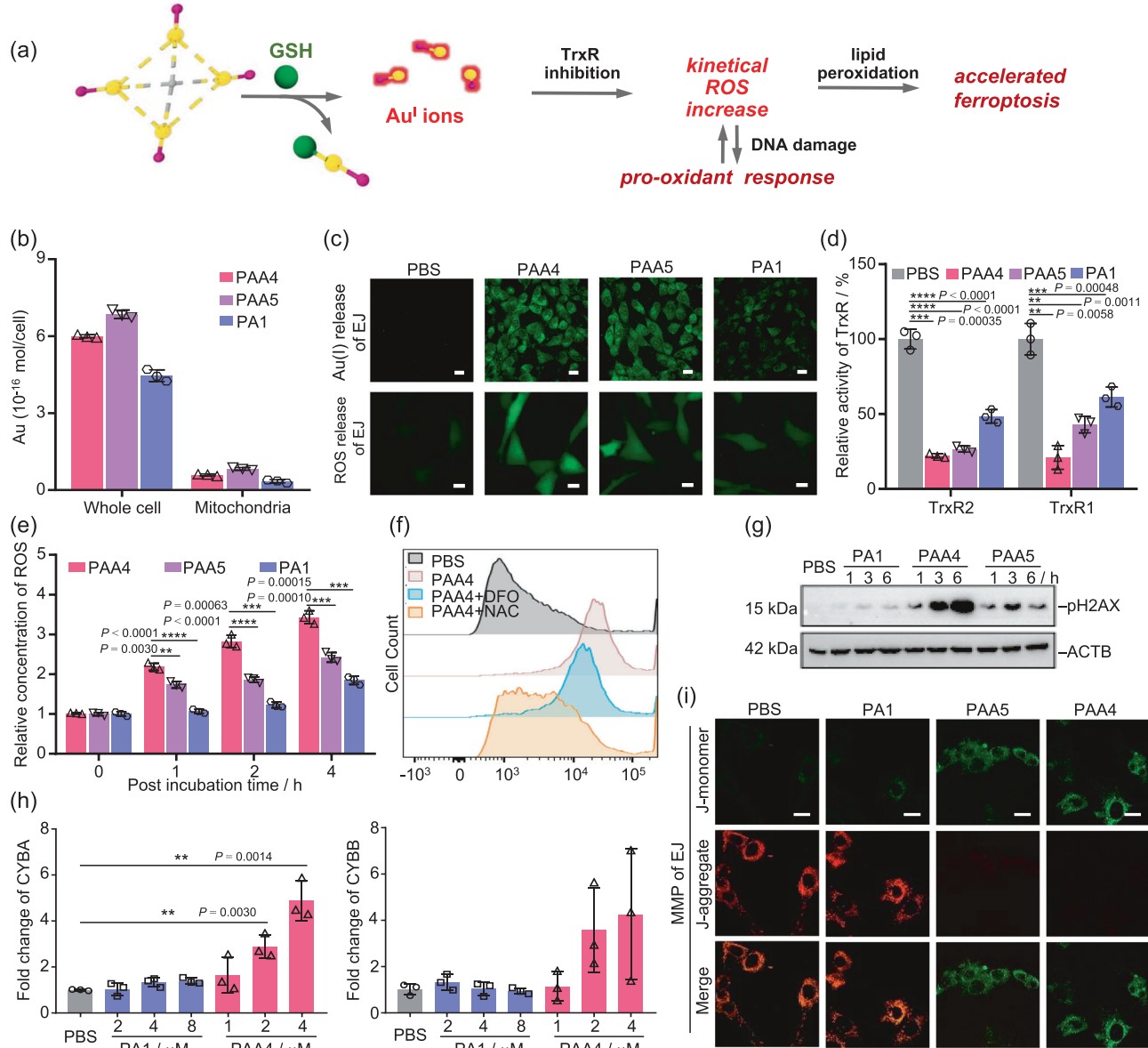

**Fig. 5 | Mechanistic studies on ferroptosis induced by PAA4 and PAA5.**
**a** Schematic pharmacological pathways of hypercarbon-centered gold(I) cluster.
**b** Content of **PAA4**, **PAA5** and **PA1** in the whole EJ cells and cell-extracted mitochondria determined by ICP-MS ($c_{\text{Au(I) clusters}} = 1.5\,\mu M$, $c_{\text{PA1}} = 6.0\,\mu M$, 4 h incubation). **c** Release of Au(I) ions monitored by confocal microscopy in the 2,3,6,7-tetrahydro-1H,5H-pyrido[3,2,1-ij]quinolin-8-yl-3-phenylpropiolate stained EJ cells, and ROS release monitored by confocal microscopy in the 2′,7′-dichlorofluorescin diacetate (DCFH-DA, 10.0 μM) stained EJ cells treated with **PAA4, PAA5** (1.5 μM, 4 h incubation) and **PA1** (6.0 μM, 4 h incubation) relative to the PBS blank trial. Scale bar: 20 mm. Excitation: 488 nm. **d** TrxR activity inhibition upon the treatment with **PAA4, PAA5** (1.5 μM, 4 h incubation) and **PA1** (6.0 μM, 4 h incubation) relative to the PBS blank trial. **e** Time-dependent intracellular ROS change in EJ cells treated by **PAA4, PAA5** (1.5 μM, 4 h incubation), and **PA1** (6.0 μM, 4 h incubation). The ROS concentration of EJ cells before treatment is referred as the control value. **f** Flow

cytometry analysis of ROS level in EJ cells treated by 2.0 μM **PAA4** with or without 2 h pretreatment of DFO (100 μM) and NAC (3.0 mM). ROS was labeled with DCFH-DA (10.0 μM). **g** Western blot analysis of pH2AX in EJ cells treated by **PA1** (8.0 μM), **PAA4** (4.0 μM) or **PAA5** (4.0 μM) for 1, 3, or 6 h, respectively. ($n = 2$ independent experiments). **h** mRNA level analysis of CYBA and CYBB in EJ cells treated by **PAA4** or **PA1** for 6 h. **i** Confocal images with the 5,5′,6,6′-tetrachloro-1,1′,3,3′-tetraethyl-imidacarbocyanine (JC-1) assay of EJ cells incubated with **PAA4, PAA5** (1.5 μM, 4 h incubation) and **PA1** (6.0 μM, 4 h incubation), respectively. Excitation: 488 and 561 nm. Scale bar: 20 mm. Asterisks (*) denote the statistical significance: $0.001 < {}^{**} P \leq 0.01$, $0.0001 < {}^{***} P \leq 0.001$, ${}^{****} P \leq 0.0001$, $P$ values were performed with one-way ANOVA followed by post hoc Tukey's test. Data were expressed as mean ± SD ($n = 3$ independent experiments examined over triplicates). Source data are provided as a Source Data file.

character of **PAA4** and **PAA5**. HUVEC cells reveals similar up-take amount of **PAA4** and **PAA5** (Supplementary Fig. S28). Subsequently, we applied a molecular sensor to monitor the active Au(I) ions released from **PAA4** and **PAA5** within EJ cells. The sensor molecule can be catalyzed by active Au(I) species to undergo a cyclization to show a green emission (Supplementary Fig. S29). As shown in Fig. 5c, remarkable green fluorescence was detected after 4 h upon the incubation with 1.5 μM **PAA4** and **PAA5**. In contrast, the cells incubated

with 6.0 μM **PA1** showed negligible fluorescence. This experiment demonstrates that **PAA4** and **PAA5** indeed release active Au(I) metabolites within cells as similar as the above on-bench experiment. We further tested the GSH level in EJ cancer cells, HUVEC normal cells, and SV-HUC-1 cells as an additional control group without overexpression of GSH. As screened by a GSH assay kit, the GSH level in the cytoplasm of EJ is 2.4 times higher than the HUVEC cell line and 3.7 times higher than the SV-HUC-1 cell line (Supplementary Fig. S30a), which is in

consistent with the cell viability results, IC50 of **PAA4** towards EJ cell line 0.8 μM, HUVEC cell line 2.1 μM and SV-HUC-1 cell line 2.3 μM (Fig. 4a and Supplementary Fig. S30b). The high GSH level and the resulting better cytotoxicity performance within EJ cells support the acceleration role of GSH in the active Au(I) release process of gold cluster prodrugs.

We also studied the inhibition activity of **PAA4** and **PAA5** towards cytosolic TrxR1 and mitochondrial TrxR2 in EJ cells, which often act as the target of gold(I) anticancer drugs due to the high affinity of gold(I) to the selenocysteine residue of TrxR[11]. The TrxR activity assay kit clearly shows the inhibition activity of both **PAA4** and **PAA5** on both TrxR1 and TrxR2 in EJ cells (Fig. 5d). Particularly, **PAA4** exhibits higher inhibition efficiency towards TrxR1 in cytoplasm than **PAA5** (TrxR1 activity in EJ cells: 21% for **PPA4** and 43% for **PAA5**), while they both have similar efficiency towards TrxR2 in mitochondria (TrxR2 activity in EJ cells: 22% for **PPA4** and 27% for **PAA5**) (Fig. 5d). These results suggest that **PAA4** is activated by GSH more promptly than **PAA5** in the transporting process from cytoplasm to mitochondria, in good agreement with the above on-bench studies. Previous studies have revealed that the inhibition of TrxR leads to the accumulation of oxidized glutathione (GSSG) and thioredoxin (Trx), and then reduces the capacity of antioxidant systems to counteract the reactive oxygen species (ROS) damage for cells[34]. Therefore, the concentrations of ROS in EJ and HUVEC cells were evaluated in vitro by a ROS assay kit (Fig. 5c, e and Supplementary Fig. S31). After 4 h incubation with **PAA4** and **PAA5**, the ROS concentration almost remains unchanged in HUVEC cells, while it is 3.4 and 2.4 times instantly enhanced in EJ cells. More importantly, the **PAA4**-induced ROS can be restrained by the ferroptosis inhibitors DFO and NAC (Fig. 5f and Supplementary Fig. S32), further supporting the ferroptosis pathway in the antitumor treatment for EJ cell. These results clearly substantiate that active Au(I) metabolites derived from the cluster prodrugs induce a kinetical ROS generation by the effective inhibition of TrxR.

The relationship between the observed kinetical increase of ROS and the accelerated ferroptosis was then deconvoluted. ROS as a signaling molecule is often generated in response to oxidative stress and participates in cell fate determination[35,36]. Generally, low or transient levels of ROS can activate survival signaling pathways and increase detoxification enzymes due to adaptive responses[37,38]. If the level of cellular ROS is too high for the intracellular antioxidant system, pro-oxidant genes will be expressed to further increase the cellular ROS level and then accelerate the PCD process[39]. In our study, we found that the excessive ROS increase as a result of the treatment of **PAA4** and **PAA5** causes significant DNA damage as indicated by the increase of histone H2AX phosphorylation (pH2AX) (Fig. 5g and Supplementary Fig. S33). Moreover, in the **PAA4**-treated EJ cells the expression of pro-oxidant genes Cytochrome b α (CYBA) and Cytochrome b β (CYBB) indeed increases 4.9 and 5.7 times, respectively, relative to the PBS group (Fig. 5h). Furthermore, no significant increase of antioxidant GSH (1.3 fold of over-expression) was observed in such intracellular pro-oxidant microenvironment treated by **PAA4** and **PAA5**, in contrast to a 2.5 fold of over-expression of GSH treated by **PA1** (Supplementary Fig. S34). The pro-oxidant response and inhibition for GSH antioxidants were further demonstrated by the changes of SLC7A11 mRNA (Supplementary Fig. S24a), which is a cystine/glutamate antiporter to implement the import of cystine for the biosynthesis of GSH and antioxidant defense[40]. In the **PAA4**- or **PAA5**-treated EJ cell samples, we first observed an upregualtion of SLC7A11 mRNA upon low dose incubation due to adaptive and controllable oxidant stress. Then, a drastic downregualtion of SLC7A11 mRNA upon high dose incubation of **PAA4** and **PAA5** occurred due to uncontrollable ROS increase. Similar changes were further found in the changes of TrxR1 mRNA (Supplementary Fig. S24b), which is agreed with the inhibition of activity of TrxR1 protein (Fig. 5d) We thus conjecture that besides the activation of pro-oxidant pathways, the deactivation of antioxidants

also accounts for the positive feedback of ROS generation. Finally, we carried out the JC-1 staining to detect mitochondrial membrane potential (MMP). As a key indicator of mitochondrial activity, high MMP represents a good capacity of electron transport and oxidative phosphorylation[41]. In healthy cells with high MMP, JC-1 forms J-aggregates with red fluorescence, while at low MMP JC-1 remains as monomers with green fluorescence. As shown in Fig. 5i, the EJ cells incubated with **PAA4** and **PAA5** give rise to green fluorescence due to the loss of MMP. We, therefore, conclude that the kinetical ROS increase derived from the pro-oxidant response is responsible for the accelerated ferroptosis within EJ cells upon the treatment by the hypercarbon-centered gold(I) cluster prodrugs.

### Antitumor effect of PAA4 and PAA5 in vivo

After verifying the effectiveness of the hypercarbon-centered gold cluster prodrugs **PAA4** and **PAA5** in vitro, we next evaluated their antitumor effect on the EJ bladder tumor in vivo. As a crucial adjuvant treatment, intravesical delivery of chemotherapeutic drugs provides a good means of selectively delivering drugs in a high concentration to the tumor-bearing bladder[42–45]. Firstly, the APBC model, in which the bladder tumor is inoculated inside an air pouch created on the back of the mouse[46,47], was established to mimic the localized bladder tumor microenvironment and test the therapeutic effect of **PAA4** and **PAA5** in vivo. As shown in magnetic resonance imaging (MRI) images (Supplementary Fig. S35), the tumors clearly appear on the inner surface of the pouch at the initial stage. In the PBS treatment control group, the tumors grow exponentially over time and show a mean volume of $1067 \pm 178$ mm$^3$ after 22 days (Supplementary Fig. S36). In contrast, **PAA4** and **PAA5** exhibit a good antitumor effect, as evidenced by the small average tumor volume of $564 \pm 180$ and $687 \pm 210$ mm$^3$ after 22 days, respectively (Supplementary Fig. S36). Meanwhile, the histological analysis on the **PAA4**- and **PAA5**-treated APBC mice together with no significant body weight loss suggest a low systemic toxicity and high biocompatibility of **PAA4** and **PAA5** (Supplementary Figs. S37 and S38). The survival time of APBC mice treated with **PAA4** (median survival time = 36 days) and **PAA5** (median survival time = 35 days) is also prolonged in comparison with the PBS control group (median survival time = 25 days) (Supplementary Fig. S39).

In order to supply more precise preclinical evaluation of the antitumor effect of **PAA4** and **PAA5**, we further built up an orthotopic bladder cancer mouse model to mimic the human counterpart (Supplementary Fig. S40). PBS, **PAA4** and **PAA5** were respectively instilled into the bladder of mice by urethral catheterization once every other day for total 5 times (Fig. 6a). Time-dependent bioluminescence images of orthotopic bladder cancer mice were obtained at different intervals after the first treatment. In contrast to a rapidly increased bioluminescence signal observed in the PBS group after 21 days, only a modest enhancement for the bioluminescence signal was observed in the **PAA4** and **PAA5** groups (Fig. 6b), substantiating the effective tumor inhibition effect (Fig. 6c). Meanwhile, the over fifty percent survivals within 50 days confirm the low systemic toxicity of **PAA4** and **PAA5** (Fig. 6d), while in contrast, the median survival time for the PBS group is 28 days. More importantly, the blood biochemistry analytic results showed no obvious difference among Saline-, **PAA4**- and **PAA5**-treated groups in the aspects of alanine aminotransferase (ALT), aspartate aminotransferase (AST), alkaline phosphatase (ALP), blood urea nitrogen (BUN) and creatinine (CRE) levels (Fig. 6e), suggesting the good biocompatibility of these metallo-prodrugs towards normal tissues in therapeutic applications. More importantly, significant PTGS2 immunofluorescence is observed in **PAA4**- and **PAA5**-treated groups, verifying the cluster-induced ferroptosis in bladder tumor. These results jointly demonstrate excellent antitumor effect and clinical translational application potential of **PAA4** and **PAA5** in bladder cancer chemotherapy by means of ferroptosis.

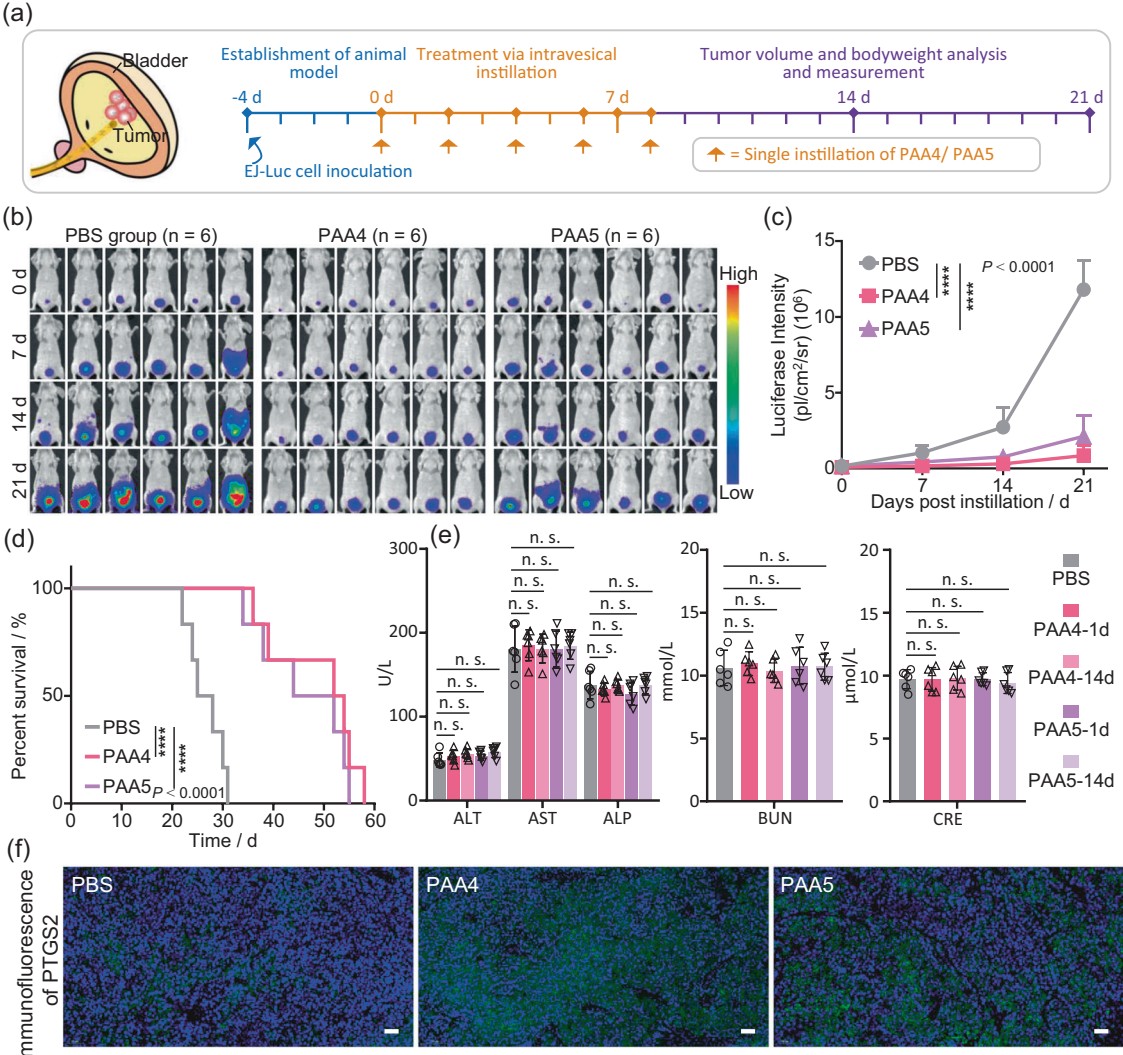

**Fig. 6 | In vivo antitumor activity evaluation of PAA4 and PAA5 in orthotopic bladder cancer mouse model. a** Left: schematic illustration of intravesical delivery of Au clusters. Right: timeline in treatment and bioluminescence imaging of orthotopic bladder cancer mice. **b** In vivo bioluminescence images of orthotopic bladder cancer mice at different days after the treatment with PBS, **PAA4** and **PAA5** (1.5 μM, 100 μL, 60 min). **c** Quantitative bioluminescence intensity of luciferase in orthotopic bladder tumor. The statistical comparison tumor growth curve was calculated on day 21. **d** Kaplan−Meier survival curve of EJ orthotopic bladder cancer nude mice. **e** Blood biochemistry data of the mice after the treatment with Saline, **PAA4** and **PAA5**, respectively, in the aspects of major indicators of liver function

including alanine aminotransferase (ALT), aspartate aminotransferase (AST), and alkaline phosphatase (ALP) levels and important indicators of kidney function including the blood urea nitrogen (BUN) and creatinine (CRE) levels. **f** Immunofluorescence analyses of PTGS2 expression in EJ tumor sections after the treatment with PBS, **PAA4** and **PAA5**. Scale bar: 40 mm. Asterisks (*) denote the statistical significance: ****$P \leq 0.0001$, $P$ values were performed with one-way ANOVA followed by post hoc Tukey's test. Data are presented as the mean ± SD ($n = 6$ mice in three independent groups from a representative experiment). Source data are provided as a Source Data file.

## Discussion

In conclusion, we develop ferroptosis-related anticancer gold(I) cluster prodrugs. The characteristic multi-centered bonding in the hypercarbon-centered cluster entities not only enables the clusters with good stability, but also facilitates a domino release of $[AuPPh_3]^+$ ions upon the activation by GSH. The better reactivity and higher cytotoxicity of two cluster compounds **PAA4** and **PAA5** relative to the mononuclear compound **PA1** in both on-bench and in vitro studies demonstrate the uniqueness of the synergistic C−Au bond dissociation. The consecutively released $[AuPPh_3]^+$ ions cause significant TrxR inhibition and kinetical ROS enhancement in cancer cells. The consequent pro-oxidant response suppresses GSH excessive upregulation and facilitates lipid peroxidation accumulation, which further elevates the ROS level in a positive feedback. The instantly induced imbalance of cellular redox environment eventually leads to an accelerated ferroptosis. This study showcases an approach to achieve ferroptosis-

based anticancer metallo-prodrugs based on the understanding of the consecutive carbon-metal bond dissociation. Further in-depth comprehension of fundamental molecular bonding in metal cluster and nanoclusters foresees a great advancement in the design and fabrication of cluster-based functional materials for biological applications.

## Methods

### General information

$^1$H, $^{13}$C, and $^{31}$P NMR were carried out on a JEOL ECX-400 MHz instrument. The UV light irradiation experiment was carried out using Agilent Cary Series UV−Vis-NIR. High resolution mass spectra were obtained on a Thermo Scientific Exactive Orbitrap instrument with an ESI mode. The mitochondria isolation kit and reactive oxygen species (ROS) assay kit were purchased from Beyotime Biotechnology Co., Ltd. (Shanghai, China). The JC-1 assay kit, TrxR activity assay kit, and glutathione (GSH) assay kit were purchased from Beijing

Solarbio Science & Technology Co., Ltd. (Beijing, China). D-Luciferin sodium salt was purchased from Shanghai Aladdin Bio-Chem Technology Co., Ltd. (Shanghai, China).EJ, HUVEC, and SV-HUC-1 cell lines were purchased from National Infrastructure of Cell Line Resource. All crystal structures were solved by direct methods, and non-hydrogen atoms were located from different Fourier maps. Non-hydrogen atoms were subjected to anisotropic refinement by full-matrix least-squares on $F^2$ using the SHELXTL program and Olex$^2$ program[48]. All pictures of crystal structures were processed by using X-seed program[49]. All animal experiments were performed in accordance with the Guide for Care and Use of Laboratory Animals, approved by the Committee for Animal Research of National Center for Nanoscience and Technology, China. License No.: SYXK 2016-0048. IACUC (Institutional Animal Care and Use Committee) Issue No. NCNST21-2202-0605. The endpoint for subcutaneous tumor xenograft was 15–20 mm at their largest dimension or maximum tumor volume of 2000 mm$^3$.

All commercially available reagents and biological assay kits were used as received. The solvents used in this study were dried by standard procedures. [O(AuPPh$_3$)$_3$]BF$_4$ (**[Au$_3$O]**)[50] and 2′-iodo-[1,1′-biphenyl]−2-amine[51] were prepared according to the published methods. The NMR spectra and ESI-MS spectra are included in Supplementary Information (Supplementary Figs. S1–S7, S41, and S42). Gating strategy is shown in Supplementary Fig. S43.

## Synthesis of 2′-((trimethylsilyl)ethynyl)-[1,1′-biphenyl]−2-amine (TEBA)

2′-Iodo-[1,1′-biphenyl]−2-amine (500 mg, 1.70 mmol), CuI (30.8 mg, 0.17 mmol) and Pd(PPh$_3$)$_2$Cl$_2$ (71.38 mg, 0.10 mmol) were dispersed in 30 mL triethylamine. The mixture was degassed by N$_2$ for three times. Then trimethylsilylacetylene (2.4 mL, 16.96 mmol) was added under inert atmosphere. The dark mixture was stirred overnight at room temperature. After removing the solvent by rotary evaporation, the residual was purified by silica gel chromatography, eluted with a mixed solution of petroleum ether and ethyl acetate (100:5 in volume). Yield: 66% (296 mg, 1.12 mmol, brown oil). $^1$H-NMR (400 MHz, DMSO-d$_6$): δ 7.55 (dd, $J$ = 8.0, 1.2 Hz, 1H), 7.47–7.42 (m, 1H), 7.37–7.31 (m, 2H), 7.05 (td, $J$ = 7.7, 1.6 Hz, 1H), 6.96 (dd, $J$ = 7.6, 1.5 Hz, 1H), 6.75 (dd, $J$ = 8.1, 0.8 Hz, 1H), 6.59 (td, $J$ = 7.4, 1.2 Hz, 1H), 4.54 (s, 2H), 0.04 (s, 9H). $^{13}$C-NMR (101 MHz, DMSO-d$_6$) δ 145.64, 143.06, 132.95, 130.94, 130.52, 129.65, 128.83, 127.73, 125.06, 122.53, 116.37, 115.51, 105.25, 97.73, 0.20. HR-MS (ESI): calcd. 266.13595 for (C$_{17}$H$_{20}$NSi)$^+$, found 266.1358.

## Synthesis of PAA4

A methanol solution (1 mL) of AgBF$_4$ (7.40 mg, 0.038 mmol) was mixed with a CH$_2$Cl$_2$ solution (1 mL) containing AuPPh$_3$Cl (18.80 mg, 0.038 mmol). The mixture was stirred at r.t. for 10 min. The supernatant was collected by centrifugation and was then added into a CH$_2$Cl$_2$ suspension of **TEBA** (10.0 mg, 0.038 mmol), **[Au$_3$O]** (56.0 mg, 0.038 mmol), KF (2.20 mg, 0.038 mmol) and K$_2$CO$_3$ (10 mg). The mixture was stirred overnight and the solution turned brown. After the filtration through diatomite and the solvent removal by rotary evaporation under vacuum, the residual was re-dissolved in CH$_2$Cl$_2$ (1 ml), which was then dropwise added into 30 mL petroleum ether under vigorous stirring. The yellow precipitate was collected by filtration. Single crystals of **PAA4** were obtained by vapor diffusion of diethyl ether into a CHCl$_3$ solution of **PAA4**. Yield: 79% (63 mg, 0.030 mmol). Yield: 79% (28 mg, 0.013 mmol). $^1$H-NMR (400 MHz, CD$_2$Cl$_2$): δ 10.47 (dd, $J$ = 8.5, 1.1 Hz, 1H), 8.55 (d, $J$ = 7.9 Hz, 1H), 8.46 (d, $J$ = 8.4 Hz, 1H), 7.80–7.68 (m, 2H), 7.62 (t, $J$ = 7.5 Hz, 1H), 7.44–7.27 (m, 38H), 7.07 (t, $J$ = 6.9, 24H). $^{31}$P-NMR (162 MHz, CD$_2$Cl$_2$) δ 31.34. HR-MS (ESI): calcd. for [**PAA4**-BF$_4$]$^+$ (C$_{86}$H$_{68}$Au$_4$NP$_4$$^+$) 2026.2959, found 2026.2942. Elemental Analysis: calcd. for (C86H68Au4BF4NP4) C, 48.86; H, 3.24; N, 0.66. Found: C, 48.88; H, 3.26; N, 0.71.

## Synthesis of PAA5

The supernatant of [(AuPPh$_3$)(BF$_4$)] (0.0094 mmol) was added into a CH$_2$Cl$_2$/methanol (v:v = 1:1) solution of **PAA4** (10.0 mg, 0.0047 mmol). The solution was stirred for 5 min. After the filtration through diatomite and the solvent removal by rotary evaporation under vacuum, the residual was re-dissolved in CH$_2$Cl$_2$ (1 ml), which was then dropwise added into 30 ml petroleum ether under vigorous stirring. The yellow precipitate was collected by filtration. Then the dissolution and precipitation were repeated one more time. Single crystals of **PAA5** were obtained by vapor diffusion of diethyl ether into a CHCl$_3$ solution of **PAA5**. Yield: 96% (12 mg, 0.0045 mmol). $^1$H-NMR (400 MHz, CD$_2$Cl$_2$): δ 10.41 (d, $J$ = 8.1 Hz, 1H), 8.54–8.43 (m, 2H), 8.12 (d, $J$ = 8.3 Hz, 1H), 7.73 (t, $J$ = 7.2 Hz, 1H), 7.60−7.44 (m, 8H), 7.30 (t, $J$ = 7.1 Hz, 17H), 7.17 (dd, $J$ = 13.0, 7.3 Hz, 23H), 7.09−7.03 (m, 2H), 6.94 (td, $J$ = 7.8, 2.2 Hz, 28H)). $^{31}$P-NMR (162 MHz, CD$_2$Cl$_2$) δ 31.10, 28.07. HR-MS (ESI): calcd. for [**PAA5**−2BF$_4$]$^{2+}$ (C$_{104}$H$_{83}$Au$_5$NP$_5$$^{2+}$) 1243.1782, found 1243.1763.

**Crystal data for PAA4 (CCDC-2003898).** C$_{92}$H$_{68}$Au$_4$BF$_4$NP$_4$O$_3$, $M$ = 2234.03, monoclinic, space group $P2/n$ (No.13), $a$ = 18.4495(7) Å, $b$ = 13.2810(4) Å, $c$ = 20.9992(7) Å, β = 115.007(4)°, $V$ = 4663.0(3) Å$^3$, $Z$ = 2, $T$ = 112.9(4) K, $D_c$ = 1.591 g cm$^{-3}$. The structure, refined on $F^2$, converged for 11125 unique reflections ($R_{int}$ = 0.0482) and 10110 observed reflections with $I$ > 2σ($I$) to give $R_1$ = 5.34% and $wR_2$ = 14.18% and a goodness-of-fit = 1.066. Due to the inclusion of highly disordered solvent molecules, there is one ALERT A about "VERY LARGE Solvent Accessible VOID(S) in Structure" in the CheckCIF report.

**Crystal data for PAA5 (CCDC-2003897).** C$_{105}$H$_{87}$Au$_5$B$_2$F$_8$NOP$_5$, $M$ = 2692.06, triclinic, space group $P$−1 (No.2), $a$ = 12.7557(4) Å, $b$ = 12.8404(3) Å, $c$ = 29.2109(9) Å, α = 102.394(2)°, β = 98.101(3)°, γ = 91.791(2)°, $V$ = 4616.9(2) Å$^3$, $Z$ = 2, $T$ = 172.99(10) K, $D_c$ = 1.936 g cm$^{-3}$. The structure, refined on $F^2$, converged for 18695 unique reflections ($R_{int}$ = 0.0618) and 14368 observed reflections with $I$ > 2σ($I$) to give $R_1$ = 6.57% and $wR_2$ = 17.34% and a goodness-of-fit = 1.100.

## X-ray crystallographic analysis

Single-crystal X-ray data for **PAA4** and **PAA5** were collected at 113 and 173 K, respectively, with Cu-Kα radiation ($l$ = 1.54178 Å) on a Rigaku Saturn 724/724+ CCD diffractometer with frames of oscillation range 0.5°. The selected crystal was mounted onto a nylon loop in polyisobutene and immersed in a low-temperature stream of dry nitrogen gas during data collection.

## Glutathione assay

The intracellular GSH levels were measured with glutathione assay kit (Solarbio Life Sciences, China). Briefly, HUVEC, EJ, and SV-HUC-1 cells at a count of $1 \times 10^6$ were seeded into six-well plates, respectively. Afterwards, total cellular proteins were obtained using RIPA buffer for testing. GSH levels were measured according to the manufacturer's instructions.

## Au(I) ions monitor

EJ cells at a density of $5 \times 10^4$ cells per well were seeded into confocal microscope dish. Then the cells were incubated with PBS, **PA1** (6.0 μM), **PAA4** (1.5 μM) and **PAA5** (1.5 μM) at 37 °C for 4 h, which were then washed with PBS for three times. Then HUVEC and EJ cells were stained with 2,3,6,7-tetrahydro-1H,5H-pyrido[3,2,1-ij]quinolin-8-yl-3-phenylpropiolate (100 μM) for another 6 h and washed with PBS for three times. Finally, confocal laser scanning microscopy (UltraVIEW VoX) was used to analyze the ROS levels.

## Reactive oxygen species assay

Intracellular ROS production was detected by Reactive Oxygen Species Assay Kit (Solarbio, CA1410) according to the manufacturer's

instructions. HUVEC and EJ cells at a density of $1 \times 10^5$ cells per well were seeded into the 96-well plates, respectively. Then the cells were incubated with PBS, **PA1** (6.0 μM), **PAA4** (1.5 μM), and **PAA5** (1.5 μM) at 37 °C for 0, 1, 2, and 4 h followed by washing with PBS for three times. The HUVEC and EJ cells were stained with 2′,7′-dichlorofluorescin diacetate (DCFH-DA, 10 μM) for another 20 min and washed with PBS for three times. Finally, microplate reader was used to analyze the ROS levels in HUVEC and EJ cells at different intervals. The relative fluorescence intensity was tested using a microplate reader (Ex = 488 nm, Em = 525 nm). Relative ROS production was calculated versus the control.

## Cell imaging assay
HUVEC and EJ cells at a density of $5 \times 10^4$ cells per well were seeded into confocal microscope dish. Then the cells were incubated with PBS, **PA1** (6.0 μM), **PAA4** (1.5 μM), and **PAA5** (1.5 μM) at 37 °C for 4 h, which were then washed with PBS for three times. Then HUVEC and EJ cells were stained with 2′,7′-dichlorofluorescin diacetate (DCFH-DA, 10 μM) for another 20 min and washed with PBS for three times. Finally, confocal laser scanning microscopy (UltraVIEW VoX) was used to analyze the ROS levels.

## Flow cytometry analysis of ROS level
EJ cells were seeded into 6 well plate at a density of $5 \times 10^5$ cells/well and cultured in the incubator overnight. DFO and NAC were added 2 h before the treatment by **PAA4** (2.0 μM) for another 4 h. Then cells were incubated with DCFH-DA (10 μM) for another 20 min and washed with PBS trice. At last, cells were harvested for flow cytometry analysis using 488 nm laser as excitation.

## Cell viability assay
The cell viability was investigated by CCK-8 assay. The cytotoxicity of phenanthridine, **PA1**, **PAA4** and **PAA5** were evaluated with HUVEC and EJ cells, respectively. Phenanthridine, **PA1**, **PAA4** and **PAA5** at different concentrations (0, 0.5, 1, 1.5, 2.0 and 4.0 μM) were prepared in cell culture medium containing 10% fetal bovine serum (FBS), 1% antibiotic solution (penicillin and streptomycin). HUVEC and EJ cells with a density of $1 \times 10^5$ cells per well were seeded into 96-well plates, respectively, then cultured at 37 °C in a humidified atmosphere with 5% $CO_2$ overnight. Then sample solutions (10 μL) at different concentrations were added to each well, which were incubated for another 24 h. Subsequently, CCK-8 solutions were added to each well and further incubated for 2 h at 37 °C. Finally, the concentration of proliferating cells in each well was measured with a microplate reader at a test wavelength of 450 nm and a reference wavelength of 690 nm. The relative cell viability rate was calculated according to the following equation:

The cell viability = $A_{samples}/A_{control} \times 100\%$. The $A_{samples}$ represent the absorbance of each group after treatment with different sample concentrations, $A_{control}$ represents the absorbance of the control group after treatment with PBS.

## Lipid peroxidation assay
EJ cells were plated into 6 well plate at a density of $5 \times 10^5$/well one day before experiment. Cells were treated with different Au compounds for 3.5 h, and then incubated with 5 μM BODIPY-C11 (ThermoFisher, D3861) at 37 °C for another 30 min. After that, cells were washed with PBS, and harvested for flow cytometry analysis using 488 nm laser as excitation. As for DFO or NAC rescue of ferroptosis, 100 μM of DFO or 3 mM of NAC were added 2 h before the treatment by Au compounds.

## Cell viability rescue assay
Rescue of cell viability by NAC or DFO was investigated by cell titer Glo assay(Promega, G7573). Briefly, EJ cells were seeded at a density of

$1 \times 10^4$ cells/well into 96 well plate and cultured at 37 °C overnight in the incubator. DFO or NAC were added to the cell culture at a final concentration of 100 μM and 3 mM, respectively. Two hours later, 4 μM of **PAA4** was added, and cells were cultured for another 12 h before addition of cell titer glo and detection of luminance using microplate reader.

## RT-qPCR analysis of mRNA expression
EJ cells were treated with different compounds for 6 h, and RNA were extracted using Trizol reagent (Life technologies), and reverse transcript was carried out using vazyme reverse transcript kit (Vazyme, R222-01). Quantitative PCR reaction was conducted using 2×SYBR green reagent (biomake, B21202) and Bio-Rad CFX 96 real-time system. Primer pairs used in this study are listed below: Actin-F: CACCA TTGGCAATGAGCGGTTC; Actin-R: AGGTCTTTGCGGATGTCCACGT; CYBA-F: ACCAGGAATTACTATGTTCGGGC; CYBA-R: TAGGTAGATG CCGCTCGCAATG; CYBB-F: CTCTGAACTTGGAGACAGGCAAA; CYBB-R: CACAGCGTGATGACAACTCCAG; PTGS2-F: CGGTGAAACTCTGGC TAGACAG; PTGS2-R: GCAAACCGTAGATGCTCAGGGA.

## ICP-MS study
EJ cells at a density of $1 \times 10^6$ cells per well were seeded into the 6-well plate and incubated overnight. **PAA4** and **PAA5** at a concentration of 1.5 μM were added and further incubated for 24 h. Then EJ cells were washed with PBS for 3 times and harvested to be centrifuged at 1200 × $g$ for 3 min and counted with cell counting chamber. Next, the cells were equally divided into 2 parts. One part was used to extract mitochondria with a mitochondria isolation kit (Beyotime). Then, both parts were transferred to MARS Vessels. After that, 1 mL of nitric acid (67%) and 3 mL of hydrochloric acid (37%) were added to these samples individually overnight. Finally, these samples were treated with 3 mL aqueous solutions containing 2% $HNO_3$ and 1% HCl. The Au standard calibration curve was acquired by analyzing a series of Au standard solutions (0.5, 1, 5, 10, and 50 ng/mL in an aqueous solution containing 2% $HNO_3$ and 1% HCl).

## Isolation of mitochondria
Mitochondria were extracted with standard differential centrifugation procedures. Firstly, EJ cells were washed three times with PBS, collected, and centrifuged at 800 × $g$ for 10 min at 4 °C. Then, the cells were added with 1 mL Lysis Buffer and homogenized with a glass homogenizer for 10 min on ice for cell lysis. The homogenate was centrifuged at 1000 × $g$ for 5 min, and the supernatant was collected and centrifuged at 12,000 × $g$ for 10 min at 4 °C. Then the precipitate was collected by discarding the supernatant and washed with Wash Buffer followed by centrifuged at 1000 × $g$ for 5 min at 4 °C. The supernatant was collected and centrifuged at 12,000 × $g$ for 10 min at 4 °C. Finally, the isolated mitochondria were obtained by discarding the supernatant and collecting the precipitate by adding 50 mL Store Buffer.

## TrxR activity assay
TrxR activity in the HUVEC and EJ cells was measured by TrxR activity assay kit after treatment with PBS, **PAA4** (1.5 μM) and **PAA5** (1.5 μM) for 4 h, respectively. Then, HUVEC and EJ cells were sonicated (300w, ultrasonic 3 s, interval 7 s, total time 3 min) in ice bath, followed by centrifuged at 10,000 × $g$, 4 °C, for 10 min. Finally, the supernatants were taken and placed on ice for testing the activity of TrxR1. Especially, the mitochondria were extracted to detect the activity of TrxR2.

## Measurement of mitochondrial membrane potential (MMP)
Mitochondrial membrane potential was measured with JC-1 assay kit for investigating mitochondria-regulated apoptosis. HUVEC and EJ cells were cultured at a density of $5 \times 10^4$ in a confocal microscope dish,

then they were incubated with PBS, **PAA4** (1.5 μM), and **PAA5** (1.5 μM) at 37 °C for 4 h, followed by washing with PBS three times and stained with JC-1 for 20 min. After removing the supernatant and washing twice with PBS buffer, the cells were immediately observed with confocal laser scanning microscopy (UltraVIEW VoX).

## Establishment and treatment of the air-pouch bladder cancer (APBC) mouse

All animal experiments were performed in accordance with the Guide for Care and Use of Laboratory Animals, approved by the Committee for Animal Research of National Center for Nanoscience and Technology, China. Female BALB/c nude mice (4–6 weeks, about 18 g) were intraperitoneally anesthetized using pentobarbital sodium (40 mg/kg body weight). Then 3–4 mL filtered sterile air was subcutaneously injected into the backs of mice to create a $3 \times 2$ cm air pouch. The volume of air pouch was preserved through injection of 1 ml sterile air on alternate days. 50 μL of the mixture of $1 \times 10^6$ EJ cells resuspended in Matrigel were seeded on the inner face of the pouch wall after 5 days. Tumor volume was measured with Vernier calipers and calculated using formula: (length × width$^2$)/2. For the treatment experiment, PBS, **PAA4** and **PAA5** (1.5 μM, 1 mL) were injected into the air-pouch and incubated for 30 min, respectively, to simulate the postoperative intravesical chemotherapy of bladder cancer.

## Establishment and treatment of orthotopic bladder cancer mouse model

Initially, female BALB/c nude mice (6–8 weeks, 16–18 g) were intraperitoneally anesthetized using pentobarbital sodium (40 mg/kg). Then a modified IV catheter (24G) was catheterized into the bladder for slightly damaging the inner mucosa of bladder. Subsequently, EJ-Luc cells were intravesical delivered into the bladder by urethral catheterization and incubated for 60 min. Next, mice were bioluminescent imaged to confirm the tumor growth in bladder. After confirming the tumor growth in bladder, PBS, **PAA4** (1.5 μM, 100 μL), and **PAA5** (1.5 μM, 100 μL) were intravesical delivered into the bladder and incubated for 60 min.

## Western blot analysis

EJ/HUVEC cells were treated with **PA1**, **PAA4**, and **PAA5** with different conditions. The lysis buffer which contains 50 mM Tris-HCl (pH 8.0), 150 mM NaCl, 1% (v/v) Triton-X 100, and protease inhibitor was used to re-suspend the pretreated cells. The estimation of the protein content was accomplished by a BCA kit (Applygen). Each sample (50 μg of protein) was subjected to SDS-PAGE and then transferred onto nitrocellulose membrane. Blots were blocked in a blocking buffer containing 5% (wt/v) non-fat milk, 0.1% (v/v) Tween-20 in 10 mM TBS, incubated overnight with primary antibodies at 4 °C on a shaker, incubated with an appropriate secondary antibody for 1 h at room temperature on a shaker and subsequently scanned on a Typhoon Trio Variable Mode Imager.

## Housing conditions for the mice

Unified temperature: 18–22 °C, lighting: 10–14 h, lights on at 8:00, lights off at 18:00 to 20:00, relative humidity: 50–60%, 6 in each cage.

## Reporting summary

Further information on research design is available in the Nature Research Reporting Summary linked to this article.

## Data availability

The X-ray crystallographic coordinates for structures reported in this article have been deposited at the Cambridge Crystallographic Data Centre (CCDC), under deposition number CCDC-2003898 (**PAA4**) and 2003897 (**PAA5**). These data can be obtained free of charge from the Cambridge Crystallographic Data Centre via www.ccdc.cam.ac.uk/data_request/cif. Source data are provided with this paper. The remaining data are available within the Article, Supplementary Information or Source Data file.

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

## Acknowledgements

Financial support by National Natural Science Foundation of China (22025105, 21772111, 21821001, 91956125, 21873079, 21573179, and 51725302) and the National Science Fund for Distinguished Young Scholars is gratefully acknowledged.

## Author contributions

L.Z., W.H., G.W., and X.W. conceived and supervised the project. K.X. carried out the synthesis and reactivity studies. N.Z., F.L., and D.H. performed the antitumor effect studies. K.X., N.Z., F.L., and D.H. co-wrote the manuscript. X.Z. performed the structural characterizations. All authors discussed the results and commented on the manuscript.

## Competing interests

The authors declare no competing interests.

## Additional information

**Supplementary information** The online version contains

supplementary material available at

