## [Peer Review File · Nature Communications]

Pro-oxidant Response and Accelerated Ferroptosis Caused by Synergetic Au(I) Release in Hypercarbon-centered Gold(I) Cluster ProdrugsREVIEWER COMMENTS

Reviewer #1 (Remarks to the Author): with expertise in gold nanoparticle

The work is original and propositive, the characterization section is clear. the composition of clusters was demonstrated by the right techniques.

I have some comments

1. Is ferroptosis a death cellular pathway different from apoptosis?
2. Caspase 3 activation and Bax pathway should be evaluated if the aim of the research group is to study the cell death mechanisms
3. what is the amount of GSH to activate the PAA4 and PAA5 clusters?
4. To conclude that PAA4 and PAA5 effectivity and their future application as alternatives for cancer treatment PAA4 and PAA5 should be evaluated using cells without overexpression of GSH
5. what is the size of a female mice urethra? the authors should provide the photographs of mice bladders with a tumor develop, bladders treated, and controls
6. How did the author determine the concentrations of PAA4 and PAA5 used in the test in vitro and in vivo?
7. What is the difference between bladder tumor microenvironment and other solid tumors?
8. In vitro test TrxR showed that PAA4 was more effective than PAA5, however, in vivo tumor growth test showed better results with PAA5. How do you explain it ?

Reviewer #2 (Remarks to the Author): with expertise in gold nanoparticle, cancer therapy

Manuscript Title: Pro-oxidant Response and Accelerated Ferroptosis Caused by Synergetic Au(I) Release in Hypercarbon-centered Gold(I) Cluster Prodrugs
Manuscript Reference: NCOMMS-21-13435A-Z

The present manuscript describes a very robust experimental plan aimed at determining whether 2 carbon-centered gold clusters, (i) a tetra-aurated complex referred to as PAA4 and (ii) a penta-aurated complex referred to as PAA5, could selectively induce cell death in human bladder cancer EJ cells due to a pro-oxidant response that would in turn accelerate ferroptosis. Additionally, the inhibition effect towards bladder tumors in vivo was evaluated with 2 distinct models with a small number of 5-6 animals per treatment group.

The experimental plan included the synthesis and characterization of the gold clusters. To induce the desired therapeutic cytotoxic effect in the cancer cells, these 2 gold clusters would rely on the GSH-triggered release of Au (I) ions, thus envisaged as pro-drugs.

The experiments proved the stability of the complexes under different conditions and additionally demonstrated that the release of Au (I) ions was more efficient in the cancer cells due to the elevated GSH contents and more acidic microenvironment relative to the non-cancer HUVEC cells used as control. These data supported the relative selectivity of the cytotoxic effects towards the cancer cells that also affords a possible explanation for the in vivo effects observed, showing low systemic toxicity but anti-tumor activity.

Regarding the stability determinations, data only for PAA4 is shown concerning the thermal stability (fig S8), in DMSO/water mixture (fig S9), in PBS at a wide range of pH (fig S10), and in DMEM (fig S11). The PAA5 counterparts should also be displayed in the supplementary data section. In figure 2, panels' c and d, the pH at which the reactions were monitored should be stated. In the text, the authors refer to a "faintly acid condition", which is an inaccurate statement of the tested experimental condition.

The subsequent in vitro experiments for the study of ferroptosis were all performed in triplicates, as described in the methods of the supplementary information section. The corresponding figure legends state that data originates from n=3. Does this mean that 3 independent experiments were performed and that each independent experiment was run in triplicates? The number of independent experiments and the number of replicates within each independent experiment should be clearly stated. Generally, for all figures (but specially in the supplementary section), data reading would benefit from more complete legends stating important experimental details such as test conditions and incubation times.

The cytotoxicity experiments indicate that PAA4 and PAA5 induce a greater cytotoxic effect towards EJ cancer cells when compared to the HUVEC. On the other hand, PA1, a mononuclear Au (I) control, was comparatively less cytotoxic towards EJ (and also HUVEC) cells. For a critical comparative analysis, the EC50 values of the cytotoxicity curves should be calculated and compared, and the drawn conclusions should be backed up with the respective appropriate statistical analysis (comparing only the EC50 values or the whole cytotoxicity curves). As an example, in the supplementary data figure S18, a significant cytotoxic effect of PA1 towards the EJ cells (and HUVEC cells also) can be depicted for the 4 μ M concentration and not only at 6 μ M as stated in the text. A statistical comparison of the cytotoxicity curves and of the EC50 would unequivocally show the differences that the authors noted but with a more accurate analysis.

The PPAR cleavage experiment is not described in the methods section of the supplementary information.

The addition of an apoptosis inhibitor, ZVAD, allegedly did not reduce the cell viability loss induced by PAA4 and PAA5. It is not clear how this confirmatory experiment was performed. In both EJ and HUVEC cells? Why was PA1 not tested under the same conditions to support the anticipated apoptotic-induced cell death in contrast with the non-apoptotic one induced by PAA4 and PAA5? Apoptotic cell death was discarded based on the results obtained with the PPAR cleavage experiments and ferroptosis was assumed based on the combination of data from increased lipid peroxidation and increased PTGS2 mRNA expression (with statistical significance only for PAA4). But why use only PTGS2 mRNA expression as a biomarker for ferroptosis? According to Chen et al, 2021: "The great challenge of PTGS2 as a biomarker of ferroptosis is that the up-regulation of PTGS2 is observed under various inflammatory conditions (FitzGerald, 2003), at least some of which are non-ferroptotic conditions." (Front. Cell Dev. Biol., <https://doi.org/10.3389/fcell.2021.637162>). Is the data presently obtained sufficient evidence of ferroptosis-induced cell death? Can other mechanisms be ruled out?

Additionally, comparative data of PTGS2 expression and lipid peroxidation (and DFO and NAC rescue) in HUVEC cells could support the selective mechanisms towards the cancer cells but data only for EJ cells are presented.

The rescuing effect of DFO and NAC in mortality of PAA4-exposed EJ cells would be more consistently demonstrated by complete cytotoxicity curves at a given time-point that could be statistically compared. At 4 μ M almost 100% mortality occurs at 24h and the cell imaging of fig. S19 indicates a very high lethality for 1.5 μ M at 4 hours. Therefore, is analyzing this DFO and NAC protective effect with 4 μ M incubations at 6h suitable?

The intracellular GSH level was found higher in EJ cells compared to HUVEC, as depicted in fig. S24. The Y axis states relative concentration of GSH, but according to the method description these should be absolute concentration values. The GSH concentration unit is not clear (neither from the method description nor from the figure S24 legend) but should be clearly indicated. Also, in the case of ROS quantification it can be assumed that relative ROS concentration means that 1 is the control value but this is not explained anywhere.

Figure S26 must be checked and if needed corrected. According to the text fig. S26 shows the reversal of ROS increase by DFO and NAC in EJ cells but the first row displays HUVEC in the legend. Why were pH2AX phosphorylation, and cytochrome b increased expressions only determined for

PAA4 and PA1? Is data lacking for PAA5?

Figure S27 shows slightly higher intracellular GSH concentrations 4h after 1.5 μ M PAA4 and PAA5 incubations and strongly increased GSH intracellular levels after 6 μ M PA1 relative to PBS incubations. The statistical analysis is lacking and should be provided. How does fig. S27 show suppression for GSH generation as stated in the text?

According to figure 4e) under the same exact experimental conditions ROS levels are greatly increased with higher ROS levels noted for PAA4 and PAA5 than with PA1. How can these conflicting data (with increasing intracellular GSH) be explained?

Figure S28 also lacks the statistical analysis that should be displayed.

The antitumor effect was investigated in vivo and tumor growth curves after treatment with either PBS (control), PAA4 and PAA% are displayed in fig. S30. Did the statistical analysis compare the whole curves as hinted by the figure inlet legend? Can these be compared with one-way ANOVA followed by post hoc Tukey's test?

The same concern applies to the Kaplan-Meier survival curves displayed in fig. S33. Wouldn't a non-parametric statistical analysis be more suited in this case?

These data analysis were also applied in the case of the second in vivo model. Therefore, the same concerns apply to the statistical analysis displayed in figure 5, on panels c and d.

After analyzing the survival curves for the APBC in vivo models it can be seen that lethality occurs in the PBS group earlier than day 24, and it is not clear when lethality starts for the other 2 groups. Nevertheless, the measurements of tumor growth and body weight were taken up to or at day 24, whereas in the case of the histology assessment of major organs the time of sampling is not indicated. Were the same number of animals tested in each case or only the surviving animals at day 24? Was the histological analysis performed for all animal at necropsy?

The same concern applies to the data displayed in figure 5 relative to the second in vivo model (panel e). The number of tested animals should be unequivocally stated as well as the respective time of the data sampling.

Reviewer #3 (Remarks to the Author): with expertise in nanomaterials, cancer therapy

In this manuscript, Zhao et al. reported a ferroptosis agent caused by Hypercarbon-centered Gold(I) cluster prodrugs. The prodrug could react with GSH and release active Au(I) to increase intracellular ROS level and decrease GSH level, therefore induce accelerated ferroptosis. The evaluation of the designed systems was well-organized and performed, the data were also well-organized. However, there are some issues that need to be solved before publication.

1. The authors mentioned several times in the manuscript that PAA4 and PAA5 showed greater toxicity in EJ cell lines compared to HUVEC cells because of higher intracellular GSH level and acidic microenvironment. Did the authors measure the GSH level of the two cell lines, or is there accurate literature to support this statement? Please measure the GSH level of the two cell lines.

2. Inhibition of SLC7A11 could reduce GSH synthesis. However, the feedback inhibition of SLC7A11 by GSH reduction reduced by PAA4 and PAA5 needs to be explained in detail. Besides, can the authors explain what is stated in the manuscript "We thus conjecture that besides the activation of pro-oxidant pathways, the deactivation of antioxidants also accounts for the positive feedback of ROS generation".

3. The authors state in the manuscript that "the UV-vis spectra remain unchanged for over hours". But as shown in Fig. S10 and S11, the authors only detected 1.0 h, so it may be imprecise. In addition, PAA4 and PAA5 appear to be a reactive substance, so the reviewer would like to know how to preserve them? Could they react with other reductive substances except GSH, after all there are many types of reducing substances in the cell?

4. Can the authors explain why the maximum value of the blue line in Fig. S15 is less than the other two groups?

5. The order of some figures in the manuscript and supporting information is different from the order mentioned in the article, such as Figure 3c and 3b, Figure S20 and S19. Please check it and adjust the original order to make the article easier to read.

6. The abbreviation for 2',7'-dichlorofluorescein diacetate is often denoted DCFH-DA, as indicated in the "Reactive oxygen species assay" section in the Supporting Information. It is abbreviated as CFDH-DA in the manuscript. Please unify it.

Reviewer #4 (Remarks to the Author): with expertise in cancer therapy

The authors describe the development and activity of Hypercarbon-centered Gold(I) Cluster Prodrugs against bladder tumours.

Whilst the paper demonstrates clear survival in vivo using a EJ-luc bladder cancer mouse model, the in vitro data regarding the mechanism of action is unclear. with respect to the role of ferroptosis. In particular, the concentrations of PAA4 and PAA5 used varies from experiment to experiment and no rationale is provided. If ferroptosis is important in the activity of these molecules, the authors need to demonstrate differences in ferroptosis marker induction at 1.5uM PAA4 and PAA5 between EJ and HUVEC cells. Specific comments are provided below.

'There is a major problem with the standard of English throughout the paper which makes it difficult to understand in places.

The numbering of the figures and figure legends needs to be corrected - scheme 1 should be Figure 1 etc.

The authors need to be specific about the time points used rather than saying for "over hours" - line 135, or "one day" line 131

This reviewer cannot find a reference to ENA in reference 21

Figure 3b uses 4uM PAA4, a dose at which there is significant loss of cell viability in HUVEC cells. The authors should aim to use doses of PAA4 or PAA5 that are not associated with significant killing of the Normal cell counterpart i.e. 1.5-2uM.

PTGS2 is only significantly induced at 4uM, not at lower doses (1-2uM) which are associated with cell death in EJ cells. The authors need to measure PTGS2 levels and rescue with DFO/NAC in HUVECs treated with 4uM PAA4 to determine whether this induction is tumour-selective. Equally, the induction of lipid peroxidation is not shown at 1.5uM and does not look to be significant at 4uM. The current data does not demonstrate the role of ferroptosis in the tumour-selective killing of EJ cells compared to HUVECs.

IN Figure 4b, the authors demonstrate fluorescence using an AU sensor molecule. These experiments demonstrate fluorescence at 1.5uM. It would be important to show that at this concentration, whether there is an increase in fluorescence in HUVECs. This would determine whether active PPA4 or PAA5 are/are not released in HUVECs and would allow the authors to better understand the tumour-selective actions of these compounds.

The authors do demonstrate that PAA4 and PAA5 induce ROS in EJ cells but not in HUVECs. However, the authors only demonstrate pH2AX induction using 4uM PAA4 or PAA5. The authors need to repeat this experiment at the lower dose of 1.5uM to demonstrate whether this is relevant to the cell killing observed.

The mouse xenograft data demonstrates a clear inhibition of tumour growth over 21 days and increased survival out to 50 days. However, it is not clear why the survival experiments were not performed beyond 50 days since there is still evidence of bioluminescence in Figure 5b indicating that there was still tumour present.

It is not clear why the detailed methods are in the supplementary information and whether this conforms to the journal guidelines.

Reviewer #5 (Remarks to the Author): with expertise in nanosystems, ferroptosis, bladder cancer models

In the present study, the authors proposed two hypercoordinated carbon-centered gold clusters, PAA4 and PAA5, and their application in cancer therapy. They found that GSH triggers the release of active Au(I) ions of PAA4 and PAA5, and the instantly massive release of coordination unsaturated Au(I) ions causes the efficient inhibition of thioredoxin reductase and then induces a rapid pro-oxidant response, consequently resulting in the occurrence of accelerated ferroptosis. The work is of great significance. However, substantial revision is still needed before its publication, and my concerns are as follows:

1. The authors demonstrate that GSH triggers the release of active Au(I) ion. What concentration of GSH is required to trigger it? How much difference in GSH content between bladder cancer cells and the normal controls? Dose the difference big enough to differently trigger the release of active Au(I) ion within the cells?
2. The results shown in Figure 2c and 2d were interpreted to be obtained under the faintly acid condition, so, what is the pH of the condition exactly?
3. Why are the ratios of the reaction system in Figure 2c (PAA4:ENA:GSH=1:0.5:1) and Figure 2d (PAA4:ENA:GSH=1:4:0.5) different? They are designed on the same purpose, or from one experiment, aren't they?
4. The authors mentioned that the acidic microenvironment contributes to the synergetic release of active Au(I) ions in bladder cancer cells, however, only stability of the drugs was assessed under different pH conditions in this study which is not strong enough to prove the contribution of acidic microenvironment to the anticancer effect of the proposed drugs specifically on cancer cells.
5. In most of the figures, 'spectra' is better to be replaced by 'spectrum'. Spectra is the plural form of spectrum.
6. How is the uptake of PAA4/5 of the normal cells? Is the relatively low cytotoxicity of PAA4 and PAA5 to normal cells due to the low uptake of PAA4 and PAA5 or the low release of active Au(I) ion after the uptake? The authors should clarify it.
7. The abbreviation of 'Oxidized glutathione' should be GSSG, not GSSH.
8. The concentrations of PAA4/5 used in the assessments differ a lot (1.5 ~ 4 μ M). Why? The concentrations of the drug generally stick to the same when assessing their biological effects.
9. More assessments or markers determination in addition to PTGS2 which was only assessed at RNA level, are needed to confirm that the proposed metallo-prodrugs exert anticancer effect via accelerating ferroptosis. Markers are better to be determined at protein level.
10. As shown in FigureS27, PAA4 and PAA5 treatment cause an elevation of GSH content in comparison to the PBS control group. However, in the manuscript, the authors demonstrate that the drugs can inhibit the generation of GSH. The interpretation is inconsistent with the result. Similarly, in Figure S28, cells treated with PAA4 and PAA5 showed upregulation of SLC7A11. How did the authors conclude the downregulation of SLC7A11 due to the treatment? Even though cells treated with high concentration of PAA4/5 showed significant higher level of GSH compared to those treated with low concentration of PAA4/5, the GSH level seems to be comparable with that in control groups.
11. The prodrugs need GSH to trigger the release of active Au(I) ions, on the other hand, the drugs inhibit the generation of GSH. Don't they contradict each other?
12. The authors conclude that the consecutively release of active Au(I) ions cause TrxR inhibition and kinetical ROS enhancement in cancer cells, and the consequent DNA damage promotes a pro-oxidant response and causes the suppression of GSH generation. The conclusion is not rigorous enough and lacks necessary evidence support, no matter from experiment or literature side. Even you observed a DNA damage and downregulation of GSH due to the treatment of the drugs, it doesn't prove the correlation or causality between them.

Response to comments

We highly appreciate the editor and reviewers for their insightful comments and helpful suggestions on our manuscript NCOMMS-21-13435A-Z. We have made revisions on the manuscript following the suggestions points by points and the details are listed below.

Reviewer #1:

1. Is ferroptosis a death cellular pathway different from apoptosis?

Response: We thank this Reviewer for his/her question. Yes, ferroptosis is a death cellular pathway different from apoptosis.

As stated in the Introduction section, ferroptosis is a new death cellular pathway different from apoptosis mainly in organelle morphology and cell metabolism aspects. The term ‘ferroptosis’ was coined in 2012 (Ref. 3 in the main text: *Cell*, **2012**, 149, 1060), defined as an iron-dependent, ROS-related form of cell death. It is associated with two main biochemical characteristics, iron accumulation and lipid peroxidation. In contrast, the apoptotic cell death is mediated by the activation of a class of proteases, such as caspase. Apoptosis is widely recognized as the programmed cell death type for mononuclear gold(I) anticancer drugs (Page 12, line 4; Ref. 9 in the main text: *Met. Ions Life Sci.*, **2018**, 18, 199). In this work, we found the occurrence of an accelerated ferroptotic cell death process induced by hypercarbon-centered gold(I) cluster prodrugs.

2. Caspase 3 activation and Bax pathway should be evaluated if the aim of the research group is to study the cell death mechanism.

Response: We thank the Reviewer for this helpful suggestion. As suggested, in this revision we assessed the caspase 3 activation and Bax pathway to study the cell death mechanism in EJ cell lysate and HUVEC cell lysate after saline, PA1, PAA4 or PAA5 treatment with western blot.

Apoptosis was previously reported as the programmed cell death type for mononuclear gold(I) anticancer drugs (*Metallomics*, **2014**, 6, 1591; *Met. Ions Life Sci.*, **2018**, 18, 199). In this work, the mononuclear compound PA1 is selected as a control group for the gold cluster compounds PAA4 and PAA5. As shown in the below Fig. A, the cleavage of caspase 3 (a key enzyme to the execution stage of an apoptotic pathway) was only detected in HUVEC cell lysate after PA1 treatment, suggesting that PA1 could induce apoptosis. Besides, the increased expression of Bax (apoptosis-promoting protein) and the decreased expression of Bcl-2 (anti-apoptosis protein) in the PA1-treated HUVEC group relative to the PBS group also support the conclusion that apoptosis is the main pathway of cell death for the PA1-treated sample.

In the case of PAA4 and PAA5, the expression of caspase 3 in both EJ and HUVEC cell lines after the treatment by PAA4 and PAA5 displays no significant decrease in comparison with the PBS group (Fig. A). Similarly, no obvious change in both Bax and Bcl-2 expression was detected in the PAA4/PAA5 treated groups. In addition to

the PARP cleavage experiment (activation biomaker of apoptosis) (Fig. B, also as Fig. S22 in the revised Supplementary Information), we conclude that PAA4 and PAA5 cannot activate the apoptosis pathway.

The western blot analysis of caspase 3 activation and Bax pathway were added in Supplementary Fig. S22 in the revised Supplementary Information. The results were added in the main text Page 12 line 5.

Fig. A Western blot analysis of Caspase 3 activation and Bax pathway in EJ cells (left) and HUVEC cells (right).

Fig. B Western blot analysis of PARP cleavage in EJ cells (left) and HUVEC cells (right).

3. what is the amount of GSH to activate the PAA4 and PAA5 clusters?

Response: We thank the Reviewer for this question. Due to more precise quantification on the amount of GSH via on-bench experiments, we applied ¹H NMR and UV-vis spectra to monitor the reaction of GSH with PAA4/PAA5. ¹H NMR studies were carried by increasing the amount of GSH to the same equivalent with PAA4/PAA5 to monitor the activation process. As shown in the on-bench ¹H-NMR monitoring experiment (Fig. A, Fig 2b in the main text), one eq. GSH is capable to completing the decomposition of one eq. PAA4 via a synergistic carbon-polymetal

bond cleavage process. For PAA5, it transforms to PAA4 first and thus needs more than two eq. GSH. UV-vis spectra were collected with GSH : PAA4/PAA5 = 50-100:1, which is similar with in vitro environment. As shown in the below Fig. B (Fig. S16 in the revised Supplementary Information), the reaction rate increases in line fitting with the amount of GSH and the reaction rate of PAA4 is higher than that of PAA5 at the same level of GSH. These results jointly demonstrate that the activation of PAA4/PAA5 clusters by GSH is a kinetically promoted process and highly concentrated GSH can accelerate the release of $[\text{AuPPh}_3]^+$ from PAA4 and PAA5.

The related discussion has been added in the revised main text Page 8, line 17.

Fig. A ^1H -NMR spectra monitoring on the reaction mixture of PAA4/GSH (left) and PAA5/GSH (right) in different ratios.

Fig. B UV-vis spectra monitoring and line fitting on the reaction mixture of PAA4/GSH (a-b) and PAA5/GSH (c-d) in different ratios.

(e) Line fitting of reaction rates.

4. To conclude that PAA4 and PAA5 effectivity and their future application as alternatives for cancer treatment PAA4 and PAA5 should be evaluated using cells without overexpression of GSH

Response: We thank the Reviewer for his/her helpful suggestion. As suggested, the

cell viability of SV-HUC-1 cell lysate, a normal urothelial cell line with much lower GSH expression than EJ cells (Fig. A Left, GSH of EJ 233.3 $\mu\text{M/gprot}$; GSH of SV-HUC-1 63.5 $\mu\text{M/gprot}$) was tested. As shown in the below Fig. A Right, PAA4/PAA5 shows higher cytotoxicity towards the GSH over-expressed EJ cells than the normal SV-HUC-1 cells (EC50 of EJ cells 0.9 μM ; EC50 of SV-HUC-1 cells 2.5 μM), indicating excellent cancer treatment potential of PAA4 and PAA5.

The results have been added in the revised manuscript (Page 13, line 25 in the main text and Fig. S16 in the revised Supplementary Information).

Fig. A Left: Intracellular GSH concentrations determined by GSH assay kits. Right: Cell viability assay of SV-HUC-1 cells treated with PAA4 and PAA5 for 24 h.

5. what is the size of a female mice urethra? the authors should provide the photographs of mice bladders with a tumor develop, bladders treated, and controls

Response: We thank the Reviewer for his/her helpful suggestion. The size of female mice (BALB/c nude) urethra is approximately 1 cm (Nature Protocols. 2009, 4, 1230-1243). Therefore, the chemotherapeutics could be directly deposited into bladder after full insertion of IV catheter (24 G, 1.9 cm). Meanwhile, the volume of a female mice (BALB/c nude) is approximately 1.5 mL, which provides enough space for mice to be treated with PAA4 and PAA5 (1 mL) by intravesical instillation.

As suggested, representative photos of mice bladders with a tumor develop, bladders treated, and controls have been provided in the revised manuscript as Fig. S39.

Fig. A Photography of mice bladders with a tumor develop, bladders treated, and controls.

6. How did the author determine the concentrations of PAA4 and PAA5 used in the test in vitro and in vivo?

Response: The in vivo treatment concentration of PAA4/PAA5 for intravesical delivery was directly instructed by 24h cell viability assay (*Angew. Chem. Int. Ed. Engl.* **2022**, e202116893). In our work, the in vivo treatment concentration 1.5 μM is above the EC50 of EJ cancer cell line but lower than the EC50 of HUVEC normal cell line (Fig. A, EC50 of EJ cells 0.9 μM (PAA4), 1.2 μM (PAA5); EC50 of HUVEC cells 2.2 μM (PAA4), 3.2 μM (PAA5)). Therefore, even under 1.5 μM distinguished treatment effect relative to the PBS group can be achieved. As to the vitro concentrations of PAA4/PAA5, 1.5-4.0 μM doses were applied and screened to monitor changes of different biomarkers within 6h treatment.

The detailed methods have been added in the revised Supplementary Information.

Fig. A Cell viability assay of EJ and HUVEC cells treated with PAA4 and PAA5 for 24 h.

7. What is the difference between bladder tumor microenvironment and other solid tumors?

Response: There are two major differences between bladder tumor and other solid tumors. First, the GSH concentration in bladder tumor is significantly higher than tumor-free bladder tissue (*Clin. Cancer Res.* **1997**, 3, 793). Furthermore, as shown in Fig. A, bladder tumor represents a readily accessible hollow organ, which provides a well-enclosed cavity for intravesical instillation instead of intravenous injection for chemotherapy of bladder cancer in the clinic (*Eur. Urol.* **2019**, 76639). Therefore, bladder tumor was selected for pro-drug therapy in our work.

Fig. A bladder tumor microenvironment

8. In vitro test TrxR showed that PAA4 was more effective than PAA5, however, in vivo tumor growth test showed better results with PAA5. How do you explain it?

Response: We are sorry for the confusion. Actually, PAA4 is always better than PAA5 in both tests. According to the in vitro test of TrxR activity (Fig. A, Fig. 4d in the main text), PAA4 exhibits higher inhibition efficiency towards TrxR1 and TrxR2 than PAA5. Meanwhile, according to the in vivo antitumor activity evaluation of PAA4 and PAA5 in orthotopic bladder cancer mouse model or air-pouch bladder cancer (APBC) model (Fig. A, Fig. S35 in Supplementary Information and Fig. 5c in the main text), PAA4 shows better antitumor effect than PAA5, which is consistent with the in vitro results.

Fig. A Left-to-Right: TrxR activity inhibition in the cellular system upon the treatment; Tumor growth curve after the treatment (APBC) model; Quantitative bioluminescence intensity of luciferase in orthotopic bladder tumor.

Reviewer #2:

1. Regarding the stability determinations, data only for PAA4 is shown concerning the thermal stability (fig S8), in DMSO/water mixture (fig S9), in PBS at a wide range of pH (fig S10), and in DMEM (fig S11). **The PAA5 counterparts should also be displayed in the supplementary data section.**

Response: We thank the Reviewer for his/her helpful suggestions. The stability of PAA5 has been monitored by ¹H-NMR as shown in Fig. S9 and S12 in the Supplementary Information. As suggested, the chemical stability of PAA5 in PBS buffer solutions with a wide range of pH (5.9-8.7) has further been validated by UV-vis monitoring in Fig. A (Fig. S13 in the revised Supplementary Information).

The stability data of PAA5 have been added in the revised Supplementary Information.

Fig. A UV-vis spectral monitoring on PAA5 in PBS butter from (a) 5.9, (b) 6.8, (c) 7.7 and (d) 8.7.

2. In figure 2, panels' c and d, the pH at which the reactions were monitored should be stated. In the text, the authors refer to a “faintly acid condition”, which is an inaccurate statement of the tested experimental condition.

Response: As suggested, the pH value was measured to be 3.6. In Fig. 2c,d, the experiments were performed in the presence of [GSH]: [HBF₄] = 1:1 (c = [1.0 × 10⁻⁴ M]). The pH value has been added in the revised manuscript.

3. The subsequent in vitro experiments for the study of ferroptosis were all performed in triplicates, as described in the methods of the supplementary information section. The corresponding figure legends state that data originates from n=3. Does this mean that 3 independent experiments were performed and that each independent experiment was run in triplicates? **The number of independent experiments and the number of replicates within each independent experiment should be clearly stated. Generally, for all figures (but specially in the supplementary section), data reading would benefit from more complete legends stating important**

experimental details such as test conditions and incubation times.

Response: We are sorry for the misleading description. As suggested, we have carefully revised the description in the legends and methods section to avoid confusions. The *in vitro* data for the study of ferroptosis were collected from three replicates in a representative experiment (independent experiment was run in triplicates) and indicated as mean \pm SD. Additionally, experimental details such as test conditions and incubation times were provided in the figure legends as suggested in the revised manuscript.

4. The cytotoxicity experiments indicate that PAA4 and PAA5 induce a greater cytotoxic effect towards EJ cancer cells when compared to the HUVEC. On the other hand, PA1, a mononuclear Au (I) control, was comparatively less cytotoxic towards EJ (and also HUVEC) cells. **For a critical comparative analysis, the EC50 values of the cytotoxicity curves should be calculated and compared, and the drawn conclusions should be backed up with the respective appropriate statistical analysis (comparing only the EC50 values or the whole cytotoxicity curves).** As an example, in the supplementary data figure S18, a significant cytotoxic effect of PA1 towards the EJ cells (and HUVEC cells also) can be depicted for the 4 μ M concentration and not only at 6 μ M as stated in the text. A statistical comparison of the cytotoxicity curves and of the EC50 would unequivocally show the differences that the authors noted but with a more accurate analysis.

Response: As suggested, we have calculated and added the EC50 values in the revised manuscript, Page 11 Fig. 3 legends; Page 14 line 3; Fig. S20 legend, (PAA4 (EC50 0.9 μ M for EJ cells and 2.2 μ M for HUVEC cells), PAA5 (EC50 1.2 μ M for EJ cells and 3.2 μ M for HUVEC cells) and PA1 (EC50 6.4 μ M for EJ cells and 7.2 μ M for HUVEC cells)).

According to the EC50 analysis, the mononuclear gold(I) compound PA1 exhibits lower cytotoxicity towards the EJ cells than that of PAA4 as shown in Fig. A below.

Fig. A Left: Cell viability assay of EJ and HUVEC cells treated with PAA4 and PAA5.

Middle: Cell viability assay of EJ cells treated with PA1.

Right: Cell viability assay of HUVEC cells treated with PA1.

5. The PARP cleavage experiment is not described in the methods section of the supplementary information.

Response: As suggested, we described the PARP cleavage experiment in the methods section of the Supplementary Information. PARP is a substrate of caspase 3. After

caspace 3 activation during apoptosis, the activated caspace 3 will cleave the 116 kDa length PARP into two parts, and the larger part of 89 kDa can be detected using PARP antibody in Western Blot experiment. The detection of PARP cleavage was performed using standard western blot procedures.

6. The addition of an apoptosis inhibitor, ZVAD, allegedly did not reduce the cell viability loss induced by PAA4 and PAA5. **It is not clear how this confirmatory experiment was performed.** In both EJ and HUVEC cells? Why was PA1 not tested under the same conditions to support the anticipated apoptotic-induced cell death in contrast with the non-apoptotic one induced by PAA4 and PAA5?

Response: The rescue experiment result by the addition of ZVAD, which is widely utilized as an apoptosis inhibitor, is shown in the below Fig. A. The ZVAD rescue experiment is applied to exclude the apoptosis mechanism for PAA4/PAA5-induced cell death of EC cancer cells. The cell death type of PA1 was validated by PARP cleavage experiments (Fig. A), which is consistent with the reported cell death type of apoptosis for mononuclear gold(I) anticancer drugs (*Metallomics*, 2014, 6, 1591).

The confirmatory experiment procedure of cell viability assay was described here for the reviewer's information. EJ cells that show lower cell viability upon PAA4/PAA5 treatment were plated in 96-well plates for overnight. Then 100 μ M ZVAD was added 2 hours before PAA4 or PAA5 treatment. After the indicated treatment durations, cell viability was analyzed using CellTiter-Glo assay (Promega, G7573). The addition of the ZVAD cannot alleviate the cell mortality in the administration samples of PAA4 and PAA5 (Fig. B), suggesting that the major cell death mechanism of PAA4 and PAA5 is not apoptosis.

Fig. A Western blot analysis of PARP cleavage in EJ cells (left) and HUVEC cells (right).

Fig. B Cell viability assay of EJ cells treated with PAA4 and PAA5 with or without ZVAD for 6 h.

7. Apoptotic cell death was discarded based on the results obtained with the PPAR cleavage experiments and ferroptosis was assumed based on the combination of data from increased lipid peroxidation and increased PTGS2 mRNA expression (with statistical significance only for PAA4). **But why use only PTGS2 mRNA expression as a biomarker for ferroptosis? According to Chen et al, 2021: “The great challenge of PTGS2 as a biomarker of ferroptosis is that the up regulation of PTGS2 is observed under various inflammatory conditions (FitzGerald, 2003), at least some of which are non-ferroptotic conditions.” (Front. Cell Dev. Biol., <https://doi.org/10.3389/fcell.2021.637162>). Is the data presently obtained sufficient evidence of ferroptosis-induced cell death? Can other mechanisms be ruled out?**

Response: We thank the Reviewer for pointing out this important issue. According to the reviewer’s suggestion, we collected more evidence to evaluate the ferroptosis-induced cell death in both mRNA level and protein level in this revision.

The cluster-induced ferroptosis was jointly demonstrated in this revised manuscript by lipid peroxidation (Fig. A, a direct biomarker for ferroptosis), up-expression of PTGS2 mRNA and ACSL4 mRNA (biomarkers for ferroptosis, *Front. Cell Dev. Biol.* 9:637162) (Fig. B), up-expression of PTGS2 protein (Fig. C), up-expression of SLC7A11 and TrxR1 mRNA (adaptive stress response at mRNA level towards ferroptosis) (Fig. B) and the rescuing effect of DFO and NAC. In addition, the well-defined ferroptosis inducer Erastin (*Cell*, 2012, 149, 1060) was applied as a positive control as shown below.

The related data have been added in the revised manuscript, referring to Fig. 3c, Fig. S23 and S24, Page 12 line 12-17 20-23, and Page 17 line 3-9.

Fig. A Flow cytometry analysis results of lipid peroxidation labeled by BODIPY-C11 in EJ cells.

Fig. B Changes of PTGS2 mRNA, SLC7A11 mRNA, TrxR1 mRNA as well as ACSL4 mRNA in EJ cells.

Fig. C Western blot analysis of PTGS2 protein up-expression in EJ cells incubated with 4.0 μ M 4h PAA4.

8. Additionally, comparative data of PTGS2 expression and lipid peroxidation (and DFO and NAC rescue) in HUVEC cells could support the selective mechanisms towards the cancer cells but data only for EJ cells are presented.

Response: As suggested, we further conducted experiments on PTGS2 protein expression (Fig. A) and lipid peroxidation (Fig. B) in HUVEC cells to support the selective mechanisms towards the cancer cells.

Upon PAA4 treatment, HUVEC cells only show negligible PTGS2 protein expression after 12 h treatment, while EJ cells express significant PTGS2 protein within 4 h. Moreover, lipid peroxidation is not detected in HUVEC cells in comparison with significant increase in EJ cells under the same condition. Therefore, we conclude that PAA4 causes selective ferroptosis towards cancer cells.

The related data of HUVEC cells have been added in the revised Supplementary Information as Fig. S23 and S26.

Fig. A Western blot analysis of PTGS2 protein expression in HUVEC cells.

Fig. B Flow cytometry analysis of lipid peroxidation labeled by BODIPY-C11 in HUVEC cells.

9. The rescuing effect of DFO and NAC in mortality of PAA4-exposed EJ cells would be more consistently demonstrated by complete cytotoxicity curves at a given time-point that could be statistically compared. At 4 μ M almost 100% mortality occurs at 24h and the cell imaging of fig. S19 indicates a very high lethality for 1.5 μ M at 4 hours. **Therefore, is analyzing this DFO and NAC protective effect with 4 μ M incubations at 6h suitable?**

Response: As suggested, cytotoxicity curves of EJ cells incubated with PAA4/PAA5 at 6h were shown below.

Fig. A Cell viability assay of EJ cells treated with PAA4 and PAA5 with or without rescue effect for 6 h.

The “4 μM incubations at 6h” condition was determined according to cell mortality results at 6h (Fig. A). Upon 4 μM 6h treatment, EJ cells show 100% mortality at 24h (Fig. B) but 73% mortality at 6h.

Fig. B Cell viability assay of EJ and HUVEC cells treated with PAA4 and PAA5 for 24 h.

DFO and NAC show protective effect with 4 μM incubations at 6h as 68% and 100% cell viability for EJ cells, respectively (Fig. A). For 1.5 μM 6h treatment, EJ cells keep ca. 100% cell viability (Fig. A).

Therefore, analyzing the DFO and NAC protective effect with 4 μM incubations at 6h is suitable.

The cell imaging in Fig. S19 (previous) has been replaced as accurate results of cell viability plots (Fig. A, new Fig. S21) for statistically comparison.

10. The intracellular GSH level was found higher in EJ cells compared to HUVEC, as depicted in fig. S24. The Y axis states relative concentration of GSH, but according to

the method description these should be absolute concentration values. **The GSH concentration unit is not clear (neither from the method description nor from the figure S24 legend) but should be clearly indicated. Also, in the case of ROS quantification it can be assumed that relative ROS concentration means that 1 is the control value but this is not explained anywhere.**

Response: The absolute concentrations of GSH were measured by reported methods. Briefly, total cellular proteins were obtained using RIPA buffer for testing. GSH levels were measured according to the manufacturer's instructions.

As suggested, we have converted the relative GSH concentration in previous figures to absolute concentration values in new Fig. S30a.

The GSH concentration in EJ and HUVEC cells is 233.28 and 62.67 $\mu\text{M/gprot}$, respectively. In addition, the relative ROS concentrations were also quantified by defining the control group in PBS without drug treatment as 1. The statement has been revised in the legends.

Intracellular GSH concentrations determined by GSH assay kits.

11. Figure S26 must be checked and if needed corrected. According to the text fig. S26 shows the reversal of ROS increase by DFO and NAC in EJ cells but the first row displays HUVEC in the legend.

Response: We are sorry for the confusion. The legend of previous Fig. S26 was incomplete. The first row of this figure (Fig. S32 in the revised Supplementary Information) illustrates the Au(I) release result in HUVEC cells. The legend has been revised as shown below.

Up: Release of Au(I) ions monitored by confocal laser scanning microscopy in the 2,3,6,7-tetrahydro-1H,5H-pyrido[3,2,1-ij]quinolin-8-yl-3-phenylpropiolate stained HUVEC cells

Middle and Down: ROS release monitored by confocal laser scanning microscopy in the 2',7'-dichlorofluorescein diacetate (DCFH-DA, 10.0 μ M) stained HUVEC cells and NAC (3.0 mM) pretreated EJ cells treated with PAA4, PAA5 (1.5 μ M, 4 h incubation) and PA1 (6.0 μ M, 4 h incubation) relative to the PBS blank trial. Scale bar: 20 μ m. Excitation: 488 nm.

12. Why were pH2AX phosphorylation, and cytochrome b increased expressions only determined for PAA4 and PA1? Is data lacking for PAA5?

Response: As suggested, western blot analysis of pH2AX in EJ cells treated by PAA5 (4.0 μ M) for 1, 3 or 6 hours have already been conducted and included in Fig. 4g in the revised main text (Fig. A below).

As to the mRNA level analysis of CYBA and CYBB, the experiments were designed to study different influence between mononuclear PA1 and clusters on the oxidant microenvironment within EJ cells. In view of the fact that PAA5 firstly transforms to PAA4 as verified by on-bench and in vitro experiments, therefore, only PAA4 was selected as a typical example for this experiment. In the PAA4-treated EJ cells, the expression of pro-oxidant genes Cytochrome b α (CYBA) and Cytochrome b β (CYBB) significantly increased. While no significant changes were observed in the PA1-treated groups. Such contrast results verify that the gold cluster PAA4 indeed induces a pro-oxidant response.

Fig. A Western blot analysis of pH2AX in EJ cells.

13. Figure S27 shows slightly higher intracellular GSH concentrations 4h after 1.5 μ M PAA4 and PAA5 incubations and strongly increased GSH intracellular levels after 6 μ M PA1 relative to PBS incubations. The statistical analysis is lacking and should be provided. **How does fig. S27 show suppression for GSH generation as stated in the text?**

Response: We thank this Reviewer for his/her question. As suggested, the statistical analysis is added as shown in Fig. A. In contrast to a 2.5 fold over-expression of GSH treated by PA1, no significant increase of antioxidant GSH was observed in such intracellular pro-oxidant microenvironment treated by PAA4 and PAA5. In general, the uptake of gold compound would promote the generation of GSH for detoxification, for example, the up-expression of GSH caused by PA1 incubation shown in Fig. A. However, the incubation of EJ cells with 4.0 μ M PAA4/PAA5 facilitates a kinetic release of Au(I), thus inducing a drastic increase of ROS. When the level of cellular ROS is too high for the intracellular antioxidant system, pro-oxidant genes would be expressed to further increase the cellular ROS level and inhibit the excessive generation of anti-oxidants GSH (*Free. Radical Bio. Med.* **2010**, *49*, 1603; *Ageing Res. Rev.* **2013**, *12*, 376).

The related discussion has been added in the new manuscript on Page 16 line 22-24.

Fig. A Intracellular GSH concentrations in EJ cells upon the treatment.

14. According to figure 4e) under the same exact experimental conditions ROS levels are greatly increased with higher ROS levels noted for PAA4 and PAA5 than with PA1. How can these conflicting data (with increasing intracellular GSH) be explained?

Response: The increase of ROS level is consistent with no significant increase of antioxidant GSH (1.3 fold of over-expression of GSH treated by PAA4 and PAA5 relative to 2.5 fold of over-expression of GSH treated by PA1). Generally, low or transient levels of ROS can activate survival signaling pathways and increase detoxification enzymes due to adaptive responses (*Free. Radical Bio. Med.* **2010**, *49*, 1603; *Ageing Res. Rev.* **2013**, *12*, 376). If the level of cellular ROS is too high for the intracellular antioxidant system, pro-oxidant genes would be expressed to further increase the cellular ROS level and then accelerate the programmed cell death process (*Redox Biol.* **2019**, *25*, 101084).

The discussion has been revised in the new manuscript accordingly Page 16 line 14-15.

15. Figure S28 also lacks the statistical analysis that should be displayed.

Response: Thanks to the reviewer's suggestion. As suggested, we have added the statistical analysis as Fig. S24 in the revised Supplementary Information.

16. The antitumor effect was investigated in vivo and tumor growth curves after treatment with either PBS (control), PAA4 and PAA% are displayed in fig. S30. **Did the statistical analysis compare the whole curves as hinted by the figure inlet legend? Can these be compared with one-way ANOVA followed by post hoc Tukey's test?**

Response: We are sorry for the confusion and thanks to the reviewer. The statistical analysis was performed by comparing the tumor volume of each mouse at Day 22 rather than the whole curves. It is compared with one-way ANOVA followed by post hoc Tukey's test.

We have carefully revised the figure legend in Fig. S35 in the revised Supplementary Information.

17. The same concern applies to the Kaplan-Meimer survival curves displayed in fig. S33. Wouldn't a non-parametric statistical analysis be more suited in this case? These data analysis was also applied in the case of the second in vivo model. **Therefore, the same concerns apply to the statistical analysis displayed in figure 5, on panels c and d.**

Response: As suggested, the Kaplan-Meimer survival curves in Fig. S38 and Fig. 5d in the revised version were analyzed with a non-parametric statistical analysis in the

revised manuscript. In Fig. S38, the statistical analysis was performed by comparing the tumor volume of each mouse at Day 21 rather than the whole curves, which is compared with one-way ANOVA followed by post hoc Tukey's test. Accordingly, we have carefully revised the figure legend in Figure 5d in the revised manuscript.

18. After analyzing the survival curves for the APBC in vivo models it can be seen that lethality occurs in the PBS group earlier than day 24, and it is not clear when lethality starts for the other 2 groups. Nevertheless, the measurements of tumor growth and body weight were taken up to or at day 24, whereas in the case of the histology assessment of major organs the time of sampling is not indicated. **Were the same number of animals tested in each case or only the surviving animals at day 24? Was the histological analysis performed for all animal at necropsy? The same concern applies to the data displayed in figure 5 relative to the second in vivo model (panel e). The number of tested animals should be unequivocally stated as well as the respective time of the data sampling.**

Response: The treatment and toxicity experiments were performed individually with different mice. The in vivo biocompatibility of PAA4 and PAA5 was investigated with health BALB/c nude mice (female, 6-8 weeks old, 16-18 g). After treatment with PBS, PAA4 and PAA5 (1.5 μ M, 100 μ L) by intravesical instillation, the mice were sacrificed to collect serum and major organs (heart, liver, spleen, lung, kidney) after 24-hour post-instillation. Subsequently, the collected major organs for the histology evaluation (Fig. S31 in the revised Supplementary Information) were performed by Wuhan Servicebio Technology Co., Ltd. The collected serum for the blood biochemistry analysis (Fig. 5e in the revised manuscript) was obtained by Hitachi Automatic Biochemical Analyzer 7100.

Reviewer #3:

1. The authors mentioned several times in the manuscript that PAA4 and PAA5 showed greater toxicity in EJ cell lines compared to HUVEC cells because of higher intracellular GSH level and acidic microenvironment. Did the authors measure the GSH level of the two cell lines, or is there accurate literature to support this statement? Please measure the GSH level of the two cell lines.

Response: We thank the Reviewer for his/her suggestions. We have carefully measured the GSH level of the EJ and HUVEC cell lines shown in Fig. S30 of Supplementary Information. As screened by a GSH assay kit shown in below Fig. A, the GSH level in the cytoplasm of EJ is 2.4 times higher than the HUVEC cell line. Moreover, Human bladder cancer cells EJ was selected because of remarkable GSH over-expression in such bladder cancer line (Ref 29, 30: *Clin. Cancer Res.* **1997**, 3, 793; *Biomarkers* **2012**, 17, 671). The high GSH level and the resulting better cytotoxicity performance within EJ cells support the key role of GSH in the active Au(I) release process of gold cluster prodrugs.

Fig. A Intracellular GSH concentrations in EJ cells upon the treatment

2. Inhibition of SLC7A11 could reduce GSH synthesis. However, the feedback inhibition of SLC7A11 by GSH reduction reduced by PAA4 and PAA5 needs to be explained in detail. Besides, can the authors explain what is stated in the manuscript "We thus conjecture that besides the activation of pro-oxidant pathways, the deactivation of antioxidants also accounts for the positive feedback of ROS generation".

Response: SLC7A11 mRNA is a cystine/glutamate antiporter to implement the import of cystine for the biosynthesis of GSH and antioxidant defense. In the PAA4- or PAA5-treated EJ cell samples, we observed an up-regulation of SLC7A11 mRNA upon low dose incubation due to adaptive and controllable oxidant stress. Then, a drastic down-regulation of SLC7A11 mRNA upon high dose incubation of PAA4 and PAA5 occurred due to an uncontrollable pro-oxidant increase (Fig. A).

Fig. A Changes of SLC7A11 mRNA in EJ cells.

Generally, low or transient levels of ROS can activate survival signaling pathways and increase detoxification enzymes due to adaptive responses (*Free. Radical Bio. Med.* **2010**, *49*, 1603; *Ageing Res. Rev.* **2013**, *12*, 376). If the level of cellular ROS is too high for the intracellular antioxidant system, pro-oxidant genes would be expressed to further increase the cellular ROS level and then accelerate the programmed cell death process via a positive feedback (*Redox Biol.* **2019**, *25*, 101084). The incubation with 4.0 μM PAA4/PAA5 facilitates a kinetic release of Au(I), thus inducing a drastic increase of ROS. Therefore, no significant increase of antioxidant GSH was observed in such intracellular pro-oxidant microenvironment treated by PAA4 and PAA5, in contrast to a 2.5 fold over-expression of GSH treated by PA1 (Fig. B), which is consistent with the changes of SLC7A11 mRNA.

The related discussion has been revised in the new manuscript accordingly, Page 16 line 14-15, 22-24 and Page 17 line 4-6.

Fig. B Intracellular GSH concentrations in EJ cells upon the treatment.

3. The authors state in the manuscript that "the UV-vis spectra remain unchanged for over hours". But as shown in Fig. S10 and S11, the authors only detected 1.0 h, so it

may be imprecise. In addition, PAA4 and PAA5 appear to be a reactive substance, so the reviewer would like to know how to preserve them? Could they react with other reductive substances except GSH, after all there are many types of reducing substances in the cell?

Response: The preservation time has been revised to precise “1.0 h” as suggested. PAA4/PAA5 are stable towards moisture and air as demonstrated by stability evaluation experiments (Fig. S8-11). They can be preserved for over six months as solid. In addition, they can be dissolved in DMSO/ethanol and dispersed in H₂O.

PAA4/PAA5 react with the thiol group of GSH via ligand exchange and protonation to release Au(I), but not by reduction. Therefore, PAA4/PAA5 are inert towards most reductive substances in the cells.

4. Can the authors explain why the maximum value of the blue line in Fig. S15 is less than the other two groups?

Response:

Fig. A UV-vis spectra monitoring on MBIA generated in the reaction mixture of PAA4, ENA and GSH.

Fig. B Reaction of probe, PAA4 and GSH.

The experiments are designed to detect the release of Au(I) from PAA4 and PAA5 upon GSH through monitoring the absorbance of the Au(I)-probe product in Fig. S15 (Fig. A). The generation rates are increased from 0.5 eq. to 2.0 eq. GSH but the maximum values are also influenced by side reaction with GSH, thus leading to lower maximum value (Fig. B).

5. The order of some figures in the manuscript and supporting information is different

from the order mentioned in the article, such as Figure 3c and 3b, Figure S20 and S19. Please check it and adjust the original order to make the article easier to read.

Response: We thank this Reviewer for his/her helpful suggestion. The order has been further checked and revised in the manuscript.

6. The abbreviation for 2',7'-dichlorofluorescein diacetate is often denoted DCFH-DA, as indicated in the "Reactive oxygen species assay" section in the Supporting Information. It is abbreviated as CFDH-DA in the manuscript. Please unify it.

Response: We thank this Reviewer for his/her helpful suggestion. The words have been corrected accordingly in the new manuscript.

Reviewer #4:

1. 'There is a major problem with the standard of English throughout the paper which makes it difficult to understand in places.

Response: We thank this Reviewer for his/her helpful suggestion. We read through the main text and corrected some spelling, tense and punctuation. The corrections were added in the revised main text.

2. The numbering of the figures and figure legends needs to be corrected - scheme 1 should be Figure 1 etc.

Response: We are sorry for the confusion and thank this Reviewer for his/her helpful suggestion. There were two "Fig. 2" mislabeled in the previous main text. And the first "Figure 2" should be the missing Figure 1. This is corrected in the revised main text.

3. The authors need to be specific about the time points used rather than saying for "over hours" - line 135, or "one day" line 131

Response: We thank this Reviewer for his/her helpful suggestion. The referred "over hours" - line 135 and "one day" line 131 have been revised as "1 h" and "24 h", respectively.

4. This reviewer cannot find a reference to ENA in reference 21

Response: ENA is the compound 4 in *Nat. Commun.* **2019**, *10*, 5639. And the synthesis of ENA can also be found in reference 21.

Synthesis of the substrate 4. 8-Indonaphthalen-1-amine (250 mg, 0.929 mmol), CuI (16.9 mg, 0.089 mmol) and Pd(PPh₃)₂Cl₂ (39.2 mg, 0.056 mmol) were dispersed in 30 mL triethylamine. The mixture in a 100-mL flask was degassed by N₂ for three times. Then trimethylsilylacetylene (1.3 mL, 9.30 mmol) was added under inert atmosphere. The dark mixture was stirred overnight at r.t. After removing the solvent by rotary evaporation, the residual was purified by silica gel column, eluted with a mixed solution of petroleum ether and ethyl acetate (100:5 in volume). Yield: 90% (200 mg, 0.836 mmol, yellow oil). ¹H-NMR (400 MHz, CD₂Cl₂): δ 7.70 (d, *J* = 7.9 Hz, 1H), 7.56–7.52 (m, 1H), 7.30–7.19 (m, 2H), 7.17–7.12 (m, 1H), 6.65 (dd, *J* = 7.4, 2.4 Hz, 1H), 5.55 (s, 2H), 0.30–0.23 (s, 9H). ¹³C-NMR (101 MHz, *d*₆-DMSO) δ 145.85, 135.63, 132.96, 131.04, 127.93, 125.17, 120.91, 117.02, 116.86, 110.55, 107.77, 100.29, 0.10. HR-MS (ESI): calcd 240.1201 for (C₁₅H₁₈NSi)⁺, found 240.1203.

5. Figure 3b uses 4uM PAA4, a dose at which there is significant loss of cell viability in HUVEC cells. The authors should aim to use doses of PAA4 or PAA5 that are not associated with significant killing of the Normal cell counterpart i.e. 1.5-2uM.

Response: Fig. 3b was tested after 6h incubation (not 24h like Fig 3a) to reveal the accelerated ferroptosis. As suggested, the cell viability was further tested after 6h incubation to reveal the difference between EJ cells and HUVEC cells by screening different doses. As shown below (Fig. A), upon incubation with 2.0 μM PAA4, EJ cells and HUVEC cells both show no significant loss of cell viability after 6h. Upon 4.0 μM incubation, EJ cells have ca. 27% cell viability after 6h while HUVEC cells still show no significant loss of cell viability. Therefore, there are no significant

killing of the HUVEC cell counterpart upon 4.0 μM incubation 6h.

The data have been summarized and added in the revised Supplementary Information as Fig. S21.

Fig. A Cell viability assay of EJ and HUVEC cells treated with PAA4 and PAA5 with or without rescue effect for 6 h.

6. PTGS2 is only significantly induced at 4 μM , not at lower doses (1-2 μM) which are associated with cell death in EJ cells. The authors need to measure PTGS2 levels and rescue with DFO/NAC in HUVECs treated with 4 μM PAA4 to determine whether this induction is tumour selective. Equally, the induction of lipid peroxidation is not shown at 1.5 μM and does not look to be significant at 4 μM . The current data does not demonstrate the role of ferroptosis in the tumor-selective killing of EJ cells compared to HUVECs.

Response: As the reviewer suggested, we further conducted experiments on mRNA expression, PTGS2 protein expression and lipid peroxidation with EJ and HUVEC cells upon 1-4 μM PAA4/PAA5 incubation to verify the tumor-selective killing effect.

EJ cells express significant PTGS2 protein at 4.0 μM 4h incubation (Fig. A), while HUVEC cells show negligible PTGS2 protein expression after 12h PAA4 treatment (Fig. B). Similarly, lipid peroxidation is significantly detected in EJ cells upon 4.0 μM 6h incubation with PAA4 in contrast to no signals in HUVEC cells upon the same treatment (Fig. C). At lower 1.5 μM dose, lipid peroxidation in EJ cells is not significant until 12h incubation (Fig. C), which is in line with the cell viability results raised in Question 5. Therefore, we conclude that PAA4 causes selective ferroptosis towards cancer cells.

The related data of HUVEC cells have been added in the revised Supplementary Information as Fig. S23 and S26.

Fig. A Western blot analysis of PTGS2 protein up-expression in EJ cells incubated with 4.0 μ M 4h PAA4.

Fig. B Western blot analysis of PTGS2 protein expression in HUVEC cells.

Fig. C Flow cytometry analysis of lipid peroxidation labeled by BODIPY-C11 in HUVEC and EJ cells.

7. In Figure 4b, the authors demonstrate fluorescence using an Au sensor molecule. These experiments demonstrate fluorescence at 1.5 μ M. It would be important to show that at this concentration, whether there is an increase in fluorescence in HUVECs. This would determine whether active PAA4 or PAA5 are/are not released in HUVECs and would allow the authors to better understand the tumor-selective actions of these compounds.

Response: We thank this Reviewer for his/her helpful suggestion. As suggested, we further monitored the release of Au(I) ions in HUVEC cells by confocal laser scanning microscopy after treated with PBS, PAA4, PAA5 (1.5 μ M, 4 h incubation)

and PA1 (6.0 μM , 4 h incubation). As shown in Fig. A, 2,3,6,7-tetrahydro-1H,5H-pyrido[3,2,1-ij]quinolin-8-yl-3-phenylpropionate stained HUVEC cells in PA1, PAA4, PAA5 group showed no obvious fluorescence enhancement compared with the PBS blank trial. This result indicates that no significant Au(I) ion release in PA1, PAA4, PAA5 treated HUVEC cells after 4h. In view of obvious Au(I) ions release in EJ cells, we draw a conclusion that PAA4 or PAA5 could be rapidly activated in EJ (tumor) cells rather than HUVEC (normal) cells.

The related data of HUVEC cells have been added in the revised Fig. S32.

Fig. A Release of Au(I) ions monitored by confocal laser scanning microscopy in the 2,3,6,7-tetrahydro-1H,5H-pyrido[3,2,1-ij]quinolin-8-yl-3-phenylpropionate stained HUVEC cells treated with PAA4, PAA5 (1.5 μM , 4 h incubation) and PA1 (6.0 μM , 4 h incubation) relative to the PBS blank trial. Scale bar: 20 μm . Excitation: 488 nm.

Fig. B (Fig. 4c in the main text) Release of Au(I) ions monitored by confocal laser scanning microscopy in EJ cells treated with PAA4, PAA5 (1.5 μM , 4 h incubation) and PA1 (6.0 μM , 4 h incubation) relative to the PBS blank trial. Scale bar: 20 μm . Excitation: 488 nm.

8. The authors do demonstrate that PAA4 and PAA5 induce ROS in EJ cells but not in HUVECs. However, the authors only demonstrate pH2AX induction using 4 μM PAA4 or PAA5. The authors need to repeat this experiment at the lower dose of 1.5 μM to demonstrate whether this is relevant to the cell killing observed.

Response: We thank this Reviewer for his/her helpful suggestion. As suggested, we further monitored time-dependent pH2AX induction upon 1.5 μM of PAA4/5 treatment in EJ and HUVEC cells (Fig. A). pH2AX is a very sensitive marker for DNA damage. Significant increase of pH2AX was found in EJ cells incubated with 1.5 μM PAA4 or PAA5 for 12h. In contrast, no pH2AX induction was observed in HUVEC cells upon 1.5 μM 24h treatment of PAA4 or PAA5.

Fig. A Western blot analysis of pH2AX in HUVEC (up) and EJ (down) cells with 1.5 μM dose of PAA4/PAA5.

9. The mouse xenograft data demonstrates a clear inhibition of tumour growth over 21 days and increased survival out to 50 days. However, it is not clear why the survival experiments were not performed beyond 50 days since there is still evidence of bioluminescence in Figure 5b indicating that there was still tumour present.

Response: Since the mice in PBS group were all dead during 22-31 days, the treatment experiments were performed until 21 days. Meanwhile, the survival experiments were performed until 60 days in our work. We illustrated the results within 50 days in the first version of our manuscript to verify the prolonged 50% survival time.

In the revised manuscript, Fig. 5b has been revised as shown below.

Kaplan-Meimer survival curve of EJ orthotopic bladder cancer nude mice.

10. It is not clear why the detailed methods are in the supplementary information and whether this conforms to the journal guidelines.

Response: We thank this Reviewer for his/her helpful suggestion. We have moved synthetic methods to the main text of the revised manuscript and sufficient detection methods were included in the supplementary information.

Reviewer #5:

1. The authors demonstrate that GSH triggers the release of active Au(I) ion. What concentration of GSH is required to trigger it? How much difference in GSH content between bladder cancer cells and the normal controls? Dose the difference big enough to differently trigger the release of active Au(I) ion within the cells?

Response: We thank this Reviewer for his/her helpful suggestions. The amounts of GSH in normal cells and in cancer cells make differences of in vitro activation of cluster pro-drugs. Due to more precise quantification via on-bench experiments, we applied ^1H NMR spectra and UV-vis spectra to monitor the reaction of GSH with PAA4/PAA5. As shown in the on-bench ^1H -NMR monitoring experiment (Fig. A, Fig 2b in the main text), one eq. GSH is capable to completing the chemical transformation of one eq. PAA4 due to a synergistic carbon-polymetal bond cleavage process. For PAA5, it transforms to PAA4 first and thus needs more than two eq. GSH. As shown in the below Fig. B (Fig. S16 in the revised Supplementary Information), the reaction rate increases in line fitting with 50-to-100 fold of GSH and the reaction rate of PAA4 is higher than that of PAA5 at the same level of GSH. These results jointly demonstrate that the activation of PAA4/PAA5 clusters by GSH is a kinetically promoted process and highly concentrated GSH can accelerate the release of Au(I) from PAA4 and PAA5. The discussion was added in the revised main text Page 8, line 17.

In this work, GSH level of EJ cells is 2.4 times higher than the HUVEC cell line (Fig. C), which is enough to discriminate different rates for the release of Au(I) ions. According to the biomarkers monitored in vitro, the release of Au(I) probe, the enhancement of ROS level, the occurrence of pro-oxidant response and ferroptosis are all detected in EJ cells.

Fig. A ^1H -NMR spectra monitoring on the reaction mixture of PAA4/GSH (left) and PAA5/GSH (right) in different ratios.

Fig. B UV-vis spectra monitoring and line fitting on the reaction mixture of PAA4/GSH (a-b) and PAA5/GSH (c-d) in different ratios. (e) Line fitting of reaction rates.

Fig. C Intracellular GSH concentrations in EJ cells upon the treatment

2. The results shown in Figure 2c and 2d were interpreted to be obtained under the faintly acid condition, so, what is the pH of the condition exactly?

Response: As suggested, the pH value was measured to be 3.6. In Fig. 2c,d, the experiments were performed in the presence of $[GSH]: [HBF_4] = 1:1$ ($c = [1.0 \times 10^{-4} M]$). The pH value has been added in the revised manuscript.

3. Why are the ratios of the reaction system in Figure 2c (PAA4: ENA:GSH=1:0.5:1) and Figure 2d (PAA4:ENA:GSH=1:4:0.5) different? They are designed on the same purpose, or from one experiment, aren't they?

Response: We thank this Reviewer for his/her question. The experiments of Figs. 2c and 2d are both designed to monitor the reaction between PAA4/PAA5 and GSH via the changes of the fluorescence and absorbance, respectively. However, in Fig. 2c, the fluorescence of ENA will be reduced at PAA4: ENA = 1:4 due to aggregate-induced

quenching. Therefore, we chose a lower ratio (PAA4: ENA = 1:0.5) as experiment conditions.

4. The authors mentioned that the acidic microenvironment contributes to the synergetic release of active Au(I) ions in bladder cancer cells, however, only stability of the drugs was assessed under different pH conditions in this study which is not strong enough to prove the contribution of acidic microenvironment to the anticancer effect of the proposed drugs specifically on cancer cells.

Response: We thank this Reviewer for his/her helpful suggestion. We supplied more evidence to prove the contribution of acidic environment to the release of cluster prodrugs.

First, in Fig. 2c, the decay of probe is promoted upon HBF_4 addition (pH = 3.6).

Second, we further directly monitor the absorbance change of the reaction between PAA4/PAA5 and GSH under the condition of $[\text{GSH}]: [\text{H}^+] = 1:1$. As monitored by the absorbance decay in the below Fig. Ab and Ad, the reaction rates of the blue lines are significantly higher in acidic environment than those red lines without HBF_4 addition, which jointly demonstrate that the acidic microenvironment could accelerate the release of Au(I).

The data have been summarized and added in the revised Supplementary Information as Fig. S16.

Fig. A UV-vis spectra monitoring and line fitting on the reaction mixture of PAA4/GSH (up) and PAA5/GSH (middle) in different ratios.

5. In most of the figures, ‘spectra’ is better to be replaced by ‘spectrum’. Spectra is the plural form of spectrum.

Response: We thank this Reviewer for his/her helpful suggestion. The words have been revised in the new manuscript.

6. How is the uptake of PAA4/5 of the normal cells? Is the relatively low cytotoxicity of PAA4 and PAA5 to normal cells due to the low uptake of PAA4 and PAA5 or the

low release of active Au(I) ion after the uptake? The authors should clarify it.

Response: As suggested, we evaluated the cellular uptake of PA1, PAA4 and PAA5 in HUVEC normal cells utilizing ICP-MS.

Compared with the uptake concentration of Au in EJ tumor cells, HUVEC cells exhibit slightly lower cellular uptake of PA1, PAA4 and PAA5 due to the slow metabolism feature of normal cells (Fig. A). However, insignificant difference of cellular uptake between HUVEC normal cells and EJ tumor cells would not induce the selective lower cytotoxicity to normal cells with a ~2.5-fold increase in EC 50 concentration (PAA4 (EC50 0.9 μ M for EJ cells and 2.2 μ M for HUVEC cells), PAA5 (EC50 1.2 μ M for EJ cells and 3.2 μ M for HUVEC cells)).

Furthermore, we investigated the release of active Au(I) ion in EJ cells and HUVEC cells by staining the Au(I) ions (Fig. B). As a result, PA1, PAA4 and PAA5 treated EJ cells showed more rapid Au(I) ion release than that of HUVEC normal cells. We hypothesize that higher GSH level in tumor cells promotes the release of active Au(I) ions from PAA4 and PAA5, and the instantly massive release of coordination unsaturated Au(I) ions causes efficient inhibition of thioredoxin reductase and then induces a rapid pro-oxidant response, consequently resulting in the occurrence of accelerated ferroptosis.

Thus, the relatively low cytotoxicity of PAA4 and PAA5 to normal cells is mainly due to the distinct release of active Au(I) ion rather than the uptake difference.

The ICP-MS data and Au(I) release data of HUVEC cells were added in the Supplementary Information as Fig. S28 and Fig. S32.

Fig. A Content of PAA4, PAA5 and PA1 in the whole HUVEC and EJ cells and cell-extracted mitochondria determined by ICP-MS.

Fig. B Release of Au(I) ions monitored by confocal laser scanning microscopy in the 2,3,6,7-tetrahydro-1H,5H-pyrido[3,2,1-ij]quinolin-8-yl-3-phenylpropiolate stained

7. The abbreviation of ‘Oxidized glutathione’ should be GSSG, not GSSH.

Response: We thank this Reviewer for his/her helpful suggestion. The words have been revised in the new manuscript.

8. The concentrations of PAA4/5 used in the assessments differ a lot (1.5 ~ 4 μM). Why? The concentrations of the drug generally stick to the same when assessing their biological effects.

Response: The clusters exhibit biological effects based on the Au(I) release, enzyme inhibition, ROS generation, pro-oxidant response and ferroptosis. Thus, we applied 1.5 ~ 4 μM 6h in vitro incubation conditions to detect the generation of biomarkers for ferroptosis.

PAA4/PAA5 is featured with the kinetic release of Au(I), thus leading to rapid generation of ROS. In a typical experiment with EJ cells, 1.5 μM 4h incubation is enough for monitoring ROS generation. However, 1.5 μM incubation does not cause significant increase of ferroptosis biomarkers until 12h, such as lipid peroxidation as shown in Fig. A below. To highlight such rapid response, we conducted experiments on ferroptosis biomarkers with 4.0 μM 6h incubation. Notably, EJ cells show ca. 27% cell viability incubated with 4.0 μM clusters 6h (Fig. B). The in vivo concentrations of 1.5 μM PAA4 and PAA5 used for tumor treatment by intravesical delivery was determined according to the results of 24h cell viability.

The data have been added in the revised Supplementary Information as Fig. S21.

Fig. A Flow cytometry analysis of lipid peroxidation labeled by BODIPY-C11 in EJ cells.

Fig. B Cell viability assay of EJ cells treated with PAA4 and PAA5 for 6 h.

9. More assessments or markers determination in addition to PTGS2 which was only assessed at RNA level, are needed to confirm that the proposed metallo-prodrugs exert anticancer effect via accelerating ferroptosis. Markers are better to be determined at protein level.

Response: As suggested, we collect more evidence to evaluate the ferroptosis-induced cell death in both mRNA level and protein level.

The cluster-induced ferroptosis is jointly demonstrated by lipid peroxidation (Fig. A, a direct biomarker for ferroptosis), up-expression of PTGS2 mRNA and ACSL4 mRNA (biomarkers for ferroptosis) (*Front. Cell Dev. Biol.* 9:637162) (Fig. B), up-expression of PTGS2 protein (Fig. C), up-expression of SLC7A11 and TrxR1 mRNA (adaptive stress response at mRNA level towards ferroptosis) (Fig. B) and the rescuing effect of DFO and NAC. In addition, the well-defined ferroptosis inducer Erastin (*Cell*, **2012**, *149*, 1060) was applied as a positive control as shown below.

The related data have been added in the revised manuscript, referring to Fig. 3c; Fig. S23, S24; Page 12 line 12-17, 20-23; Page 17 line 3-9.

Fig. A Flow cytometry analysis results of lipid peroxidation labeled by BODIPY-C11 in EJ cells.

Fig. B Changes of PTGS2 mRNA, SLC7A11 mRNA, TrxR1 mRNA as well as ACSL4 mRNA in EJ cells.

Fig. C Western blot analysis of PTGS2 protein up-expression in EJ cells incubated

with 4.0 μM 4h PAA4.

10. As shown in FigureS27, PAA4 and PAA5 treatment cause an elevation of GSH content in comparison to the PBS control group. However, in the manuscript, the authors demonstrate that the drugs can inhibit the generation of GSH. The interpretation is inconsistent with the result. Similarly, in Figure S28, cells treated with PAA4 and PAA5 showed upregulation of SLC7A11. How did the authors conclude the downregulation of SLC7A11 due to the treatment? Even though cells treated with high concentration of PAA4/5 showed significant higher level of GSH compared to those treated with low concentration of PAA4/5, the GSH level seems to be comparable with that in control groups.

Response:

Fig. A Intracellular GSH concentrations in EJ cells upon the treatment.

In Fig. S27 (Fig. A), no significant increase of antioxidant GSH was observed in such intracellular pro-oxidant microenvironment treated by PAA4 and PAA5, in contrast to a 2.5 fold of over-expression of GSH treated by PA1.

In general, the uptake of gold compound will directly promote the generation of GSH for detoxification, for example, the up-expression of GSH caused by PA1 incubation shown in Fig. A. However, the incubation with 4.0 μM PAA4/PAA5 facilitates a kinetic release of Au(I), thus inducing a drastic increase of ROS. If the level of cellular ROS is too high for the intracellular antioxidant system, pro-oxidant genes will be expressed to further increase the cellular ROS level and inhibit the excessive generation of anti-oxidants GSH (*Free. Radical Bio. Med.* **2010**, *49*, 1603; *Ageing Res. Rev.* **2013**, *12*, 376).

Fig. B Changes of SLC7A11 mRNA in EJ cells.

There is an up-regulation of SLC7A11 mRNA upon low dose incubation due to adaptive and controllable oxidant stress. Then, a drastic down-regulation of SLC7A11 mRNA upon high dose incubation was observed due to uncontrollable pro-oxidant increase Fig. S28 (Fig. B). Such down-regulation of SLC7A11 mRNA is in consistence with no significant increase of antioxidant GSH.

SLC7A11 mRNA is a cystine/glutamate antiporter to implement the import of cystine for the biosynthesis of GSH and antioxidant defense. In the PAA4- or PAA5-treated EJ cell samples, we first observed an up-regulation of SLC7A11 mRNA upon low dose incubation due to adaptive and controllable oxidant stress. Then, a drastic down-regulation of SLC7A11 mRNA upon high dose incubation of PAA4 and PAA5 occurred due to uncontrollable pro-oxidant increase (Fig. B).

The sentences have been revised in the new manuscript accordingly, Page 16 line 22-24; Page 17 line 4-6.

11. The prodrugs need GSH to trigger the release of active Au(I) ions, on the other hand, the drugs inhibit the generation of GSH. Don't they contradict each other?

Response:

Once prodrugs taken by the cells, current high GSH concentration would promote the rapid Au(I) release, then Au(I) ions continue to bind with target enzymes. In the second step, excessive generation of GSH due to adaptive stress competes with target enzymes to bind with Au(I) ions, leading to detoxification. Therefore, the inhibition of

excessive generation of GSH by drastic pro-oxidant response after prodrug release would further promote the following Au(I) ions binding with target enzyme and then increase pro-oxidant environment in positive feedback. Such virtuous circle of promotion rapidly increases oxidant levels in cells, which can be evidenced by DNA damage and MMP loss. Eventually, ferroptosis pathway is activated and cell death irreversibly occurs.

In a word, abundant GSH will firstly trigger the release of cluster prodrugs and the subsequent pro-oxidant response will inhibit the excessive increase of GSH, promoting the generated active Au(I) to binding with target enzyme in positive feedback.

12. The authors conclude that the consecutively release of active Au(I) ions cause TrxR inhibition and kinetical ROS enhancement in cancer cells, and the consequent DNA damage promotes a pro-oxidant response and causes the suppression of GSH generation. The conclusion is not rigorous enough and lacks necessary evidence support, no matter from experiment or literature side. Even you observed a DNA damage and downregulation of GSH due to the treatment of the drugs, it doesn't prove the correlation or causality between them.

Response: We thank this Reviewer for his/her helpful suggestion. As suggested, the descriptions were revised as shown below.

The mechanism of inhibition of TrxR by Au(I) ions is well established by previous works (*Coord. Chem. Rev.* **2009**, 253, 1692). The inhibition of TrxR leads to the accumulation of oxidized glutathione (GSSH) and thioredoxin (Trx), and then reduces the capacity of antioxidant systems to counteract the reactive oxygen species (ROS) damage for cells (*Mol. Nutr. Food Res.* **2009**, 53, 87). Then drastic ROS enhancement is accompanied with pro-oxidant response in positive feedback, which jointly inhibit the multi-antioxidant systems within cells. Such inhibition is reported to be crucial for lipid peroxidation accumulation and the resulting ferroptosis. DNA damage, suppression for GSH excessive generation and MMP loss are just the consequences of ROS enhancement and pro-oxidant response. They do not directly involve within the ferroptosis pathway.

Thus, we revise the conclusion to remove "DNA damage and down-regulation of GSH" in the new manuscript as suggested.

REVIEWERS' COMMENTS

Reviewer #1 (Remarks to the Author):

all comments were attended by the authors and included in the text, I recommend to include the bladder photographs in the final document just to enrich the paper.

Reviewer #2 (Remarks to the Author):

Manuscript Title: Pro-oxidant Response and Accelerated Ferroptosis Caused by Synergetic Au(I) Release in Hypercarbon-centered Gold(I) Cluster Prodrugs
Manuscript Reference: NCOMMS-21-13435B

After reading the rebuttal to my comments and to those of the other reviewers it is my opinion that the authors made a significant effort to address all comments. They have made considerable changes to the original manuscript and generally addressed the reviewer's concerns adequately.

My concerns regarding this revised version are listed below:

1-A few grammar incorrections can be noted. This revised version of the manuscript should be again carefully revised for grammar and style; e.g. (among many others) systematic toxicity would be systemic toxicity;

2-The authors confirmed that only one single independent in vitro experiment was performed (run in triplicates). This means that the results should be interpreted with caution as a minimum number of three independent experiments with distinct cells is recommended. This should be mentioned and discussed as a limitation to the data;

3-The statistical comparisons of whole cytotoxicity curves and of the corresponding EC50 are still lacking and would be much more accurate and informative;

4-In many cases I could not find in the revised manuscript or in the revised supplementary material, the changes that were mentioned in the rebuttal letter; specially the additional experimental details that should be included in the figure legends seem to be still lacking (the manuscript should be revised to include this information); As an example, in figures S35, S38 and 5d it is still not clear to me that the statistical comparison was made at days 22 or 21. Neither the figure caption nor the legend state this clearly;

5-Regarding the in vivo data, the authors mention on the rebuttal letter that the "biocompatibility experiments" comprising the histological analysis and serum biomarkers were determined "24-hour post-instillation". Maybe this was a mistake and the time point was not at 24h. In this case, please state the correct time-point (24 days?). Otherwise, the data obtained is irrelevant.

Reviewer #3 (Remarks to the Author):

The authors have addressed my main concerns. I have no further recommendations.

Reviewer #5 (Remarks to the Author):

The authors have answered all my questions and I don't have further question for the authors.

We thank the editor and reviewers for their insightful and helpful suggestions and have made the following changes to the manuscript.

Reviewer #1:

1. I recommend to include the bladder photographs in the final document just to enrich the paper..

Response: We thank the Reviewer for his/her helpful suggestion. It has been added in SI as Fig. S20.

Reviewer #2:

1. A few grammar incorrections can be noted. This revised version of the manuscript should be again carefully revised for grammar and style; e.g. (among many others) systematic toxicity would be systemic toxicity;

Response: We thank the Reviewer for his/her helpful suggestion. We have carefully checked and revised the manuscript.

2. The authors confirmed that only one single independent in vitro experiment was performed (run in triplicates). This means that the results should be interpreted with caution as a minimum number of three independent experiments with distinct cells is recommended. This should be mentioned and discussed as a limitation to the data;

Response: We thank the Reviewer for his/her helpful suggestion. Every data point was examined over three representative experiments (three independent experiments in three cells for in vitro experiments) in triplicates (three replicates in a representative experiment). Accordingly, the legends have been revised as “n = 3 independent experiments examined over triplicates; n = 6 animals examined over triplicates”.

3. The statistical comparisons of whole cytotoxicity curves and of the corresponding EC50 are still lacking and would be much more accurate and informative;

Response: We thank the Reviewer for his/her helpful suggestion. We have calculated the IC50 (half maximal inhibitory concentration) of PAA4 to EJ cells (0.80 μ M), HUVEC cells (2.1 μ M) and SV-HUC-1 cells (2.3 μ M), PAA5 to EJ cells (1.1 μ M), HUVEC cells (2.7 μ M) and SV-HUC-1 cells (2.3 μ M), PA1 (IC50 6.0 μ M for EJ cells and 7.3 μ M for HUVEC cells). The data was presented in Fig. 4a, Fig. S20 and Fig. S30b.

4. In many cases I could not find in the revised manuscript or in the revised supplementary material, the changes that were mentioned in the rebuttal letter; specially the additional experimental details that should be included in the figure legends seem to be still lacking (the manuscript should be revised to include this information); As an example, in figures S35, S38 and 5d it is still not clear to me that the statistical comparison was made at days 22 or 21. Neither the figure caption nor the legend state this clearly;

Response: We thank the Reviewer for his/her helpful suggestion. We carefully checked the figure legends and added experimental details in the revised manuscript

or in the revised supplementary material. In details, the statistical comparison tumor growth curve presented in Fig. S36 (APBC model) was calculated on day 22. In Fig. S38, the body weight changes (APBC model) were monitored every two days for 22 days. Quantitative bioluminescence intensity of luciferase in orthotopic bladder tumor was calculated on day 22 in Fig. 5c. The Kaplan-Meier survival curve comparison analysis of EJ orthotopic bladder cancer nude mice presented in Fig. 5d was performed by Log-rank (Mantel-Cox) test through GraphPad prism 8.0. The above-mentioned experimental details were supplemented in the captions.

5. Regarding the in vivo data, the authors mention on the rebuttal letter that the “biocompatibility experiments” comprising the histological analysis and serum biomarkers were determined “24-hour post-instillation”. Maybe this was a mistake and the time point was not at 24h. In this case, please state the correct time-point (24 days?). Otherwise, the data obtained is irrelevant. I recommend to include the bladder photographs in the final document just to enrich the paper..

Response: We thank the Reviewer for his/her helpful suggestion. In the blood biochemistry data presented in the previous manuscript, the mice were sacrificed to collect serum and major organs (heart, liver, spleen, lung, kidney) after 24-hour intravesical installation PAA4 and PAA5 (1.5 μ M, 100 μ L) to evaluate the acute toxicity. We also performed a blood biochemistry test 14 days post-installation to further confirmed the biocompatibility according to the standard acute toxicity evaluation cycle.